# Targeting mutant RAS in patient-derived colorectal cancer organoids by combinatorial drug screening

Carla S Verissimo[1,2†], René M Overmeer[1,2†], Bas Ponsioen[1,2†], Jarno Drost[2,3], Sander Mertens[1,2], Ingrid Verlaan-Klink[1,2], Bastiaan van Gerwen[4], Marieke van der Ven[4], Marc van de Wetering[2,3], David A Egan[5], René Bernards[2,6], Hans Clevers[2,3], Johannes L Bos[1,2], Hugo J Snippert[1,2*]

[1]Molecular Cancer Research, Center for Molecular Medicine, University Medical Center Utrecht, Utrecht, Netherlands; [2]Cancer Genomics Netherlands, Utrecht, Netherlands; [3]Hubrecht Institute – KNAW, University Medical Center Utrecht, Utrecht, The Netherlands; [4]Mouse Clinic for Cancer and Aging, Netherlands Cancer Institute, Amsterdam, The Netherlands; [5]Cell Biology, Center for Molecular Medicine, University Medical Center Utrecht, Utrecht, The Netherlands; [6]Division of Molecular Carcinogenesis, Netherlands Cancer Institute, Amsterdam, The Netherlands

*For correspondence: h.j.g. snippert@umcutrecht.nl

[†]These authors contributed equally to this work

**Abstract** Colorectal cancer (CRC) organoids can be derived from almost all CRC patients and therefore capture the genetic diversity of this disease. We assembled a panel of CRC organoids carrying either wild-type or mutant RAS, as well as normal organoids and tumor organoids with a CRISPR-introduced oncogenic *KRAS* mutation. Using this panel, we evaluated RAS pathway inhibitors and drug combinations that are currently in clinical trial for RAS mutant cancers. Presence of mutant RAS correlated strongly with resistance to these targeted therapies. This was observed in tumorigenic as well as in normal organoids. Moreover, dual inhibition of the EGFR-MEK-ERK pathway in RAS mutant organoids induced a transient cell-cycle arrest rather than cell death. In vivo drug response of xenotransplanted RAS mutant organoids confirmed this growth arrest upon pan-HER/MEK combination therapy. Altogether, our studies demonstrate the potential of patient-derived CRC organoid libraries in evaluating inhibitors and drug combinations in a preclinical setting.

## Introduction

One of the great challenges in targeted cancer treatment has been the development of effective RAS-targeting drugs. RAS mutations occur in about 15% of all human tumors (*Bos, 1989*) and so far all attempts to selectively interfere in mutant RAS signaling have failed in the clinic (*Stephen et al., 2014*; *Cox et al., 2014*). Progress has long been impeded by the fact that the currently used model systems to pre-test drugs are insufficient: cell lines, on the one hand, have very limited genetic diversity, while mouse models on the other hand, may not represent human tumors (*Sachs and Clevers, 2014*; *Gould et al., 2015*). Moreover, until recently, personalized medicine required large-scale in-vitro screening on short-term cultures of tumor sections (*Centenera et al., 2013*), or alternatively, resource-intensive in-vivo screens using xenotransplantation of tumors into immunodeficient mice (*Jin et al., 2010*; *Tentler et al., 2012*). Recently, stem-cell based organoid technology was introduced to establish long-term cultures of both normal and tumor tissues from various organs (*Sato et al., 2009*, *2011*; *Bartfeld et al., 2015*; *Boj et al., 2015*; *Huch et al., 2015*; *Karthaus et al.,*

**eLife digest** Recent technical advances mean that miniature replicas of many tissues can be grown in the laboratory. These so-called organoids provide scientists with model systems that are not as limited as simple, two-dimensional sheets of cells growing in a petri dish, and less labor and resource intensive than studies using laboratory animals. In particular, organoids grown from tumor cells from cancer patients have been suggested as having numerous advantages over both laboratory-grown cancer cells and mice when it comes to testing potential new anticancer drugs.

Mutations in a gene called *KRAS* are common in many types of cancer including colon cancer. Tumors with these mutations are difficult to treat and so far virtually all attempts to generate compounds that selectively interfere with the KRAS protein encoded by the mutant gene have failed. Instead, drugs that indirectly inhibit this protein's effects by targeting other proteins in the same signaling pathway are currently being tested on patients. However, there is still a need for better ways to pre-test whether these drugs will be effective in humans without having to expose the patient to side effects or an ineffective drug.

Now, Verissimo, Overmeer, Ponsioen et al. have tested clinically-used KRAS pathway inhibitors and drug combinations against normal colon organoids and colon cancer organoids derived from patients with colon cancer. Gene editing techniques were used to introduce *KRAS* mutations into some of the normal organoids grown from healthy tissue, and into cancer organoids grown from tumors that had a normal copy of the *KRAS* gene. In all cases, only those organoids with mutant forms of the *KRAS* gene were resistant to the treatments. Furthermore, when organoids with the *KRAS* mutation were treated with some combination therapies that are currently being tested in clinical trials, the tumors stopped growing but the tumor cells failed to die. Similar drug treatments on mice carrying human colon cancer organoids confirmed these results, which is in line with previous studies where tumor tissue from human patients was transplanted into mice.

These findings show that collections of tumor organoids from multiple patients could help researchers to quickly identify and optimize targeted anticancer therapies before they are incorporated into clinical trials. In the future, clinical studies are needed to verify how accurately the testing of cancer drugs on organoids predicts whether the drug will or will not work in patients.

*2014*; *Gao et al., 2014*). The advantage of this technology is that it can capture the genetic diversity of both normal and tumor tissues. Indeed, for colorectal cancer (CRC) a genetically diverse Biobank of patient-derived CRC organoids was established and used to integrate genomic data and mono-therapy drug responses at the level of individual patient-derived organoid lines (*van de Wetering et al., 2015*).

We employed this biobank to further explore potential strategies to target mutant RAS, including the combination therapy of pan-HER and MEK inhibition, which is currently tested in clinical trials. We confirm the strong correlation between the presence of mutant RAS and resistance towards EGFR inhibition. Our data reinforce the notion that an oncogenic mutation in *RAS* is sufficient to confer this resistance independent of cellular status, whether it concerns normal or tumorigenic cells. Moreover, real-time imaging of the resistant drug response at the cellular level reveals predominant cell-cycle arrest in RAS mutant organoids, in contrast with the complete induction of cell death in CRC organoids with WT RAS. In vivo drug response of xenotransplanted RAS mutant CRC organoids confirmed the arrest in tumor growth upon dual inhibition of the EGFR-MEK-ERK pathway. Finally, efficient inhibition by dual targeting of the mutant RAS pathway strongly sensitizes for the induction of cell death, as illustrated by minimal addition of BCL inhibition. Our studies demonstrate the strong potential of patient-derived CRC organoid libraries in evaluating inhibitors and drug combinations in a preclinical setting.

## Results

### Drug response of patient-derived CRC organoids with and without mutant KRAS

To explore drug responses of patient-derived CRC organoids towards combination therapies of targeted inhibitors of the EGFR-RAS-ERK pathway, we applied a drug sensitivity screen using EGFR-family and MEK inhibitors (EGFRi and MEKi resp.) either as mono or combination therapy on two cancer organoids from a previously established biobank of CRC organoids (*van de Wetering et al., 2015*). To start, we chose cancer organoids from the individuals P8 and P26, which share a similar composition of frequent cancer mutations such as functionally inactive *APC* and *TP53*. However, they differ in their KRAS status. P8T contains wild-type (WT) *KRAS*, while P26T contains an oncogenic mutant version of *KRAS* (G12V).

3D-organoids were challenged with drugs for 72 hr and drug responses were determined by quantifying cell viability through measurements of ATP levels using CellTiter-Glo (*van de Wetering et al., 2015*). We observed the expected sensitivity of P8T towards afatinib (irreversible EGFR/HER2 inhibitor) and insensitivity of KRAS mutant P26T (*Figure 1A*). Selumetinib (MEKi) as a monotherapy showed little efficacy in both P8T and P26T, but combination therapy confirmed previous findings that MEKi sensitizes RAS mutant tumor cells to EGFR/HER2 inhibition (*Figure 1A*) (*Sun et al., 2014*). However, the KRAS mutant P26T organoids were less sensitive to the combination therapy than the KRAS WT P8T organoids.

To monitor drug response on a cellular level, we stably introduced DNA constructs encoding fluorescently-labeled H2B and performed real-time confocal imaging on the 3D-organoids for 72 hr in the presence and absence of drugs. We performed EGFR-RAS-ERK pathway inhibition with relatively high concentrations of afatinib (1 μM) in combination with selumetinib (1 μM). In P26T (mutant KRAS) we only observed cell cycle arrest with very limited cell death induction. This was in stark contrast with the very rapid induction of cell death in P8T (WT KRAS) (*Figure 1B*, *Video 1*). When we repeated these imaging experiments using much lower drug concentrations, we noticed a general shift to resistance for both organoid lines. Under these conditions, also P8T predominantly showed cell cycle arrest rather than cell death, and the cancer cells in P26T organoids even continued to proliferate (*Figure 1—figure supplement 1*, *Video 2*). Taken together, our data indicate that 72 hr of combination treatment with afatinib and selumetinib (EGFRi/HER2i and MEKi) effectively kills KRAS WT P8T organoids, while the mutant KRAS P26T organoids are significantly less sensitive.

### In vivo drug response of xenotranplanted patient-derived cancer organoids

In order to validate the observed drug response of in vitro cultured organoids in an in vivo model, we xenotransplanted P18T and P26T tumor organoids in immunodeficient mice. In line with a previous report where only engineered tumor progression organoids with increasing number of cancer mutations (APC, KRAS, P53 and/or SMAD4) showed efficient engraftment (*Drost et al., 2015*), we only obtained reliable engraftment using P26T CRC organoids. We initially started using concentration schedules of afatinib and selumetinib that had previously been reported (*Sun et al., 2014*), but we observed no significant effect of the drug combination on tumor growth over time (*Figure 2A*). To exclude that the tumors had acquired resistance during the in vivo drug treatment, we isolated the tumors to re-establish secondary organoids and subjected these to identical drug tests. Dose-response curves on these secondary organoids were identical to the parental organoid line P26T, independent of the type of drug treatment that the tumors underwent in the mice (*Figure 2—figure supplement 1*). Indeed, in agreement with lower drug concentrations that proved to be ineffective in blocking proliferation in vitro (*Figure 1—figure supplement 1*, *Video 1*), we speculate that the in vivo drug concentrations were insufficient to effectively block the EGFR-MEK-ERK pathway. To confirm this hypothesis, we further increased the drug levels to high but tolerable doses. This indeed induced significant growth stabilization (but no regression) of P26T xenotransplanted tumor in mice (*Figure 2B*), in agreement with loss of proliferative activity as was also detected in vitro (*Figure 1B*). The fact that in vivo xenografted CRC organoids yields similar drug responses as in vitro organoid cultures and identical to previous reported drug response of KRAS mutant PDX models of CRC (*Sun et al., 2014*), validates the testing and evaluation of targeted inhibitors in CRC organoids.

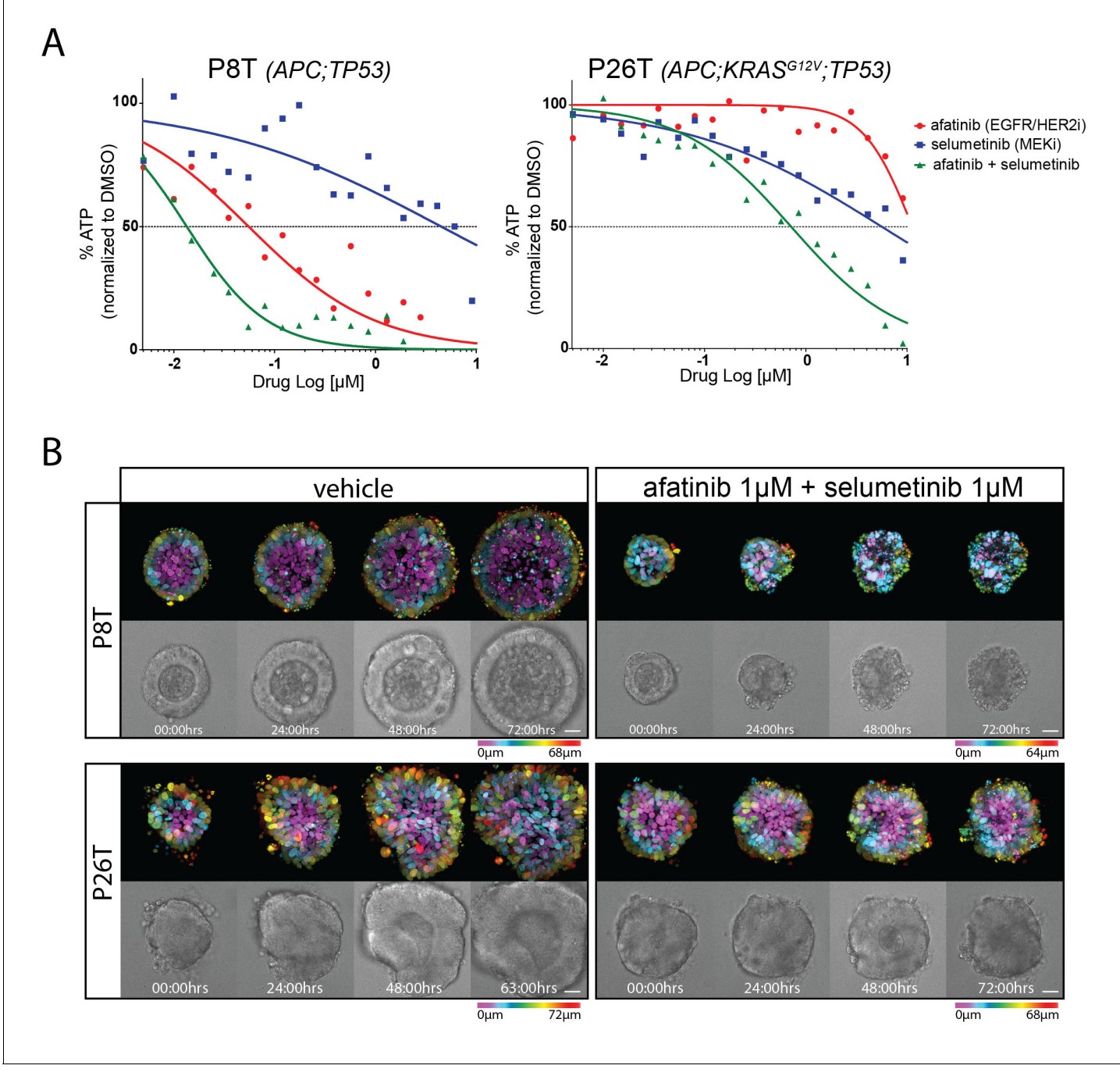

**Figure 1.** Drug responses of patient-derived CRC organoids with and without mutant KRAS. (**A**) Dose-response curves of patient-derived CRC organoids P8T (KRAS[WT]; APC and TP53 mutant) and P26T (KRAS[G12V]; APC and TP53 mutant) treated with the dual EGFR/HER2 inhibitor afatinib, MEK inhibitor selumetinib or a combination thereof. Cell viability was measured by an ATP-based assay after 72 hr of drug treatment. (**B**) Stills from representative time-lapse imaging (three days) of CRC organoids P8T and P26T treated with vehicle (DMSO) or a combination of targeted inhibitors afatinib and selumetinib (both 1 μM) (see also *Video 1*). In every panel, upper images show color-coded depth of maximum-projected z-stacks of H2B-mNeonGreen fluorescent organoids. Lower panels: corresponding transmitted light images. Time interval: 15 min. Scale bars: 20 μm. Representative time-lapse of two experiments is shown (total six organoids/condition).

The following source data and figure supplement are available for figure 1:

**Source data 1.** ImageJ/Fiji macro script: 'Organoid movie macro'.

*Figure 1 continued on next page*

*Figure 1 continued*

**Figure supplement 1.** Stills from representative time-lapse imaging (three days) of CRC organoids P8T and P26T treated with vehicle (DMSO) or a combination of targeted inhibitors afatinib (33 nM) and selumetinib (200 nM) (see also *Video 2*).

# CRISPR genome-editing in CRC organoids reveals profound effect of KRAS$^{G12D}$ on drug response

P8T and P26T CRCs are microsatellite-stable (MSS) and belong to the same molecular subtype classification based on RNA expression data (TA, also referred to as canonical CMS2 according to consensus classification) (*van de Wetering et al., 2015*; *Guinney et al., 2015*). Genomic characterization of these patient-derived CRC organoids in comparison to their matched normal tissue revealed many additional mutations within the protein coding sequence of the genome (*van de Wetering et al., 2015*). For P8T and P26T, 230 and 163 of such cancer specific mutations were detected respectively (*van de Wetering et al., 2015*). To exclude potential contributions of all these additional mutations to the effect that oncogenic KRAS imposes on drug responses, we introduced an oncogenic *KRAS* mutation in patient-derived CRC organoid P18T via CRISPR/Cas9-mediated homologous recombination (*Drost et al., 2015*). Like P8T, original P18T is WT for the entire downstream EGFR signaling pathway. P18T-KRAS$^{G12D}$ mutant cells were generated as reported previously for normal colon organoids (*Drost et al., 2015*) and genotyping of clonally expanded organoids confirmed that the clones contained the *KRAS$^{G12D}$* mutation (*Figure 3A*), as well as a Cas9-mediated inactivation of the second allele by introducing an 86 bp deletion. Upon addition of oncogenic KRAS, no overall differences in morphology or growth rates were observed during normal culture conditions.

To investigate the exclusive effect of oncogenic KRAS on a combination therapy that targets the EGFR-RAS-ERK pathway, we performed a full matrix screen of 14 drug concentrations over a 5 nM to 5 µM range of both the targeted inhibitors afatinib (EGFR/HER2i) and selumetinib (MEKi) (*Figure 3B*). Notably, their combined administration is currently used in a clinical trial for patients with *RAS* mutant CRCs (NCT02450656). While the original P18T demonstrated high sensitivity to EGFR/HER2 inhibition by monotherapy, a single introduced oncogenic point mutation in *KRAS*

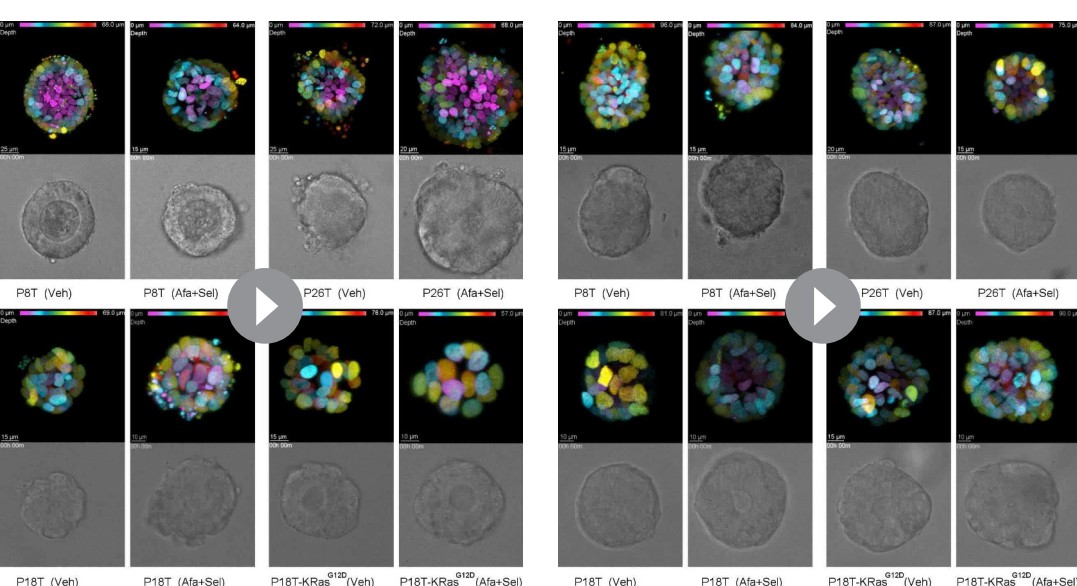

**Video 1.** Real-time imaging of cellular drug responses in tumor organoids using high concentrations targeted inhibitors.

**Video 2.** Real-time imaging of cellular drug responses in tumor organoids using low concentrations targeted inhibitors.

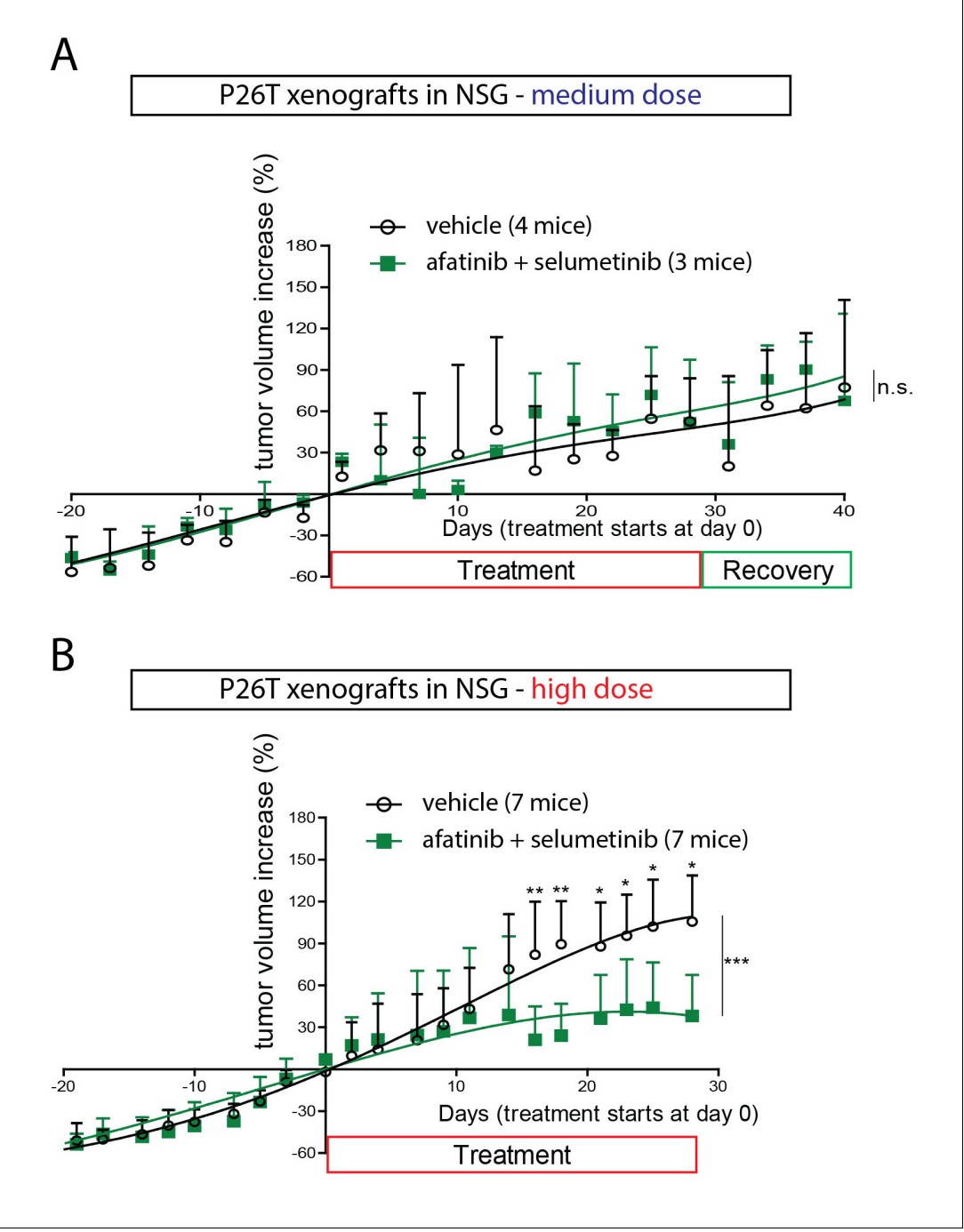

**Figure 2.** In vivo drug response of xenotransplanted CRC organoids. (**A**) P26T CRC organoids were subcutaneously transplanted in immunodeficient mice. Once tumors have grown to a volume of 300 mm$^3$, animals were treated for 28 days with vehicle, afatinib (12,5 mg/kg; five days on, two days off), selumetinib (20 mg/kg; five days on, two days off) or both drugs in combination. The mean percentage change in tumor volume relative to initial tumor volume is shown. Error bars represent standard deviation. n.s., not significant. (**B**) Same experimental setup as in **A**, but with increased drug concentrations for afatinib (20 mg/kg; five days on, two days off) and selumetinib (25 mg/kg; five days on, two days off); as well as in combined treatment. Error bars represent standard deviation. *p<0,05; **p<0,01; ***p<0001.

The following figure supplement is available for figure 2:

*Figure 2 continued on next page*

*Figure 2 continued*

**Figure supplement 1.** Secondary organoid cultures were derived from xenografted P26T tumors (organoid-derived xenograft, ODX) of mice that have been treated with vehicle, afatinib (12,5 mg/kg; five days on, two days off), selumetinib (20 mg/kg; five days on, two days off) or both.

provided resistance to EGFR/HER2 inhibition. Moreover, we analyzed combination effects using the Bliss independence model. Positive Bliss scores indicate combinatorial effects that exceed additive effects. The heat map of Bliss scores for P18T and P18T-KRAS shows that a large range of concentrations for both compounds show positive scores, but that presence of oncogenic KRAS renders the loss of viability and positive Bliss range towards higher drug concentrations indicating resistance (*Figure 3—figure supplement 1*).

Next, we again studied the cellular drug response by real-time imaging. Reminiscent of the patient-derived CRC organoid with an endogenous *RAS* mutation (P26T), we noticed that the introduction of oncogenic KRAS renders a CRC organoid less sensitive to the afatinib/selumetinib combination therapy (*Figure 3C*, *Video 1*). More specifically, quantifications of all mitotic and apoptotic events during the filmed drug response revealed both loss of proliferation and apoptosis induction in P18T, while P18T-KRAS$^{G12D}$ only showed reduced proliferation but unchanged apoptosis rates (*Figure 3D* and *Figure 3—figure supplement 2*).

Despite the phenotypic difference in drug response, pERK levels in both tumor organoids were severely reduced (*Figure 3—figure supplement 3*). Since suboptimal suppression of ERK activity might permit tumor growth in BRAF mutant cancers (*Bollag et al., 2010*; *Corcoran et al., 2015*), we determined the cellular effects of drug response when lowering drug concentrations. Since significant differential effects were observed between P18T and P18T-KRAS$^{G12D}$ during the matrix screen around 33 nM afatinib + 200 nM selumetinib (*Figure 3B*), we repeated real-time imaging of drug response using these lower drug concentrations. As with P8T and P26T, we noticed a general shift from sensitivity towards resistance when drug concentrations were reduced. More specifically, the RAS WT cancer organoids showed cell cycle arrest rather than cell death, while the RAS mutant organoids appeared unaffected and sustained proliferation (*Figure 3—figure supplement 4*, *Video 2*).

## Differential drug sensitivity in CRC organoids with and without mutant RAS upon combination therapies that include EGFR inhibition

Considering the isogenic CRC organoids P18T and P18T-KRAS$^{G12D}$ as our gold standard to reveal the specific effects of KRAS$^{G12D}$ on drug responses, we expanded our focus at targeting the linear EGFR-RAS-ERK pathway with the ultimate aim to find a targeted therapy that is specifically effective against RAS mutant CRCs. Multiple targeted inhibitors against identical targets were used to exclude artifacts and to increase the mechanistic significance behind the rationale of potential therapies (*Figure 4A*; and *Figure 4—source data 1* and *Supplementary file 1* for all dose-response curves) of which few combination therapies are in clinical trial (*Figure 4B*).

First, we noticed a much lower sensitivity of P18T-KRAS$^{G12D}$ for pan-HER inhibitors afatinib, lapatinib and dacomitinib in contrast to the parental P18T (much lower IC$_{50}$, *Figure 3—figure supplement 1*). Second, within P18T hardly any additive sensitivity could be observed when EGFRi was complemented with MEK or ERK inhibition (*Figure 4—figure supplement 1*). In contrast, dual-targeting strategies strongly enhanced efficacies in P18T-KRAS$^{G12D}$ regardless which specific inhibitor combination was used (*Figure 4—figure supplement 1*). Nevertheless, all tested combinations that included EGFR inhibition revealed stronger negative effect on cellular viability in P18T than in P18T with mutant KRAS (positive ΔIC$_{50}$'s, *Figure 4—figure supplement 1*). In contrast, most mono- and combination therapies against MEK and/or ERK that excluded EGFRi showed on average similar efficacies in P18T-KRAS$^{G12D}$ as in P18T (*Figure 4—figure supplement 1*).

In parallel, we tested dual-targeting strategies involving PI3K-AKT and EGFR-RAS-ERK pathways considering the interconnectivity between these pathways (*Figure 4—figure supplement 2A*). Like MEK or ERK inhibition, we observed that pharmacological inhibition of PI3K or AKT in combination with anti-EGFR therapy did not enhance efficacy in a KRAS mutant background (*Figure 4—figure supplement 2B*). In line with this, clinical studies focusing on combining MEK inhibitors with PI3K,

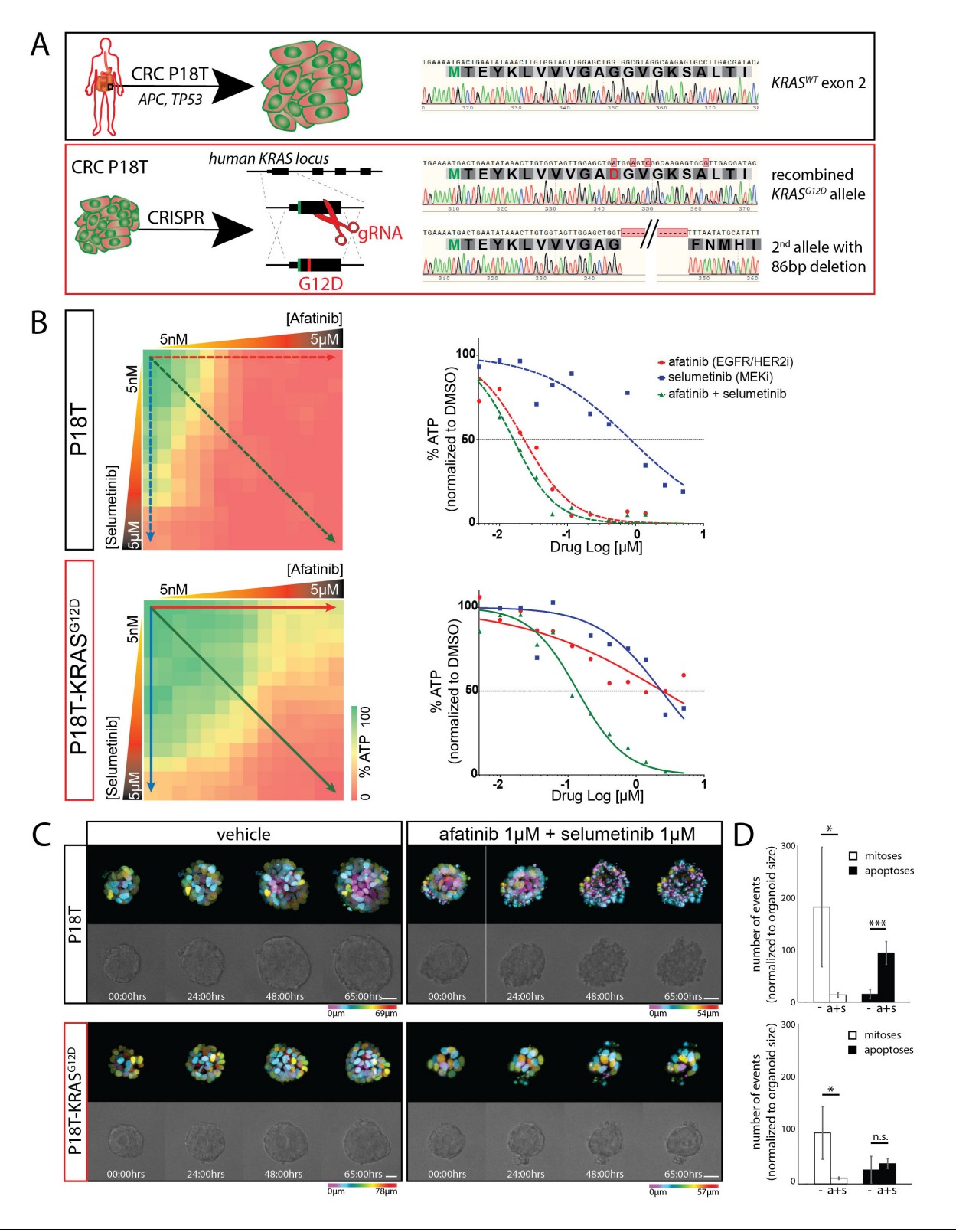

**Figure 3.** CRISPR genome editing in CRC organoids reveals effect of KRAS[G12D] on drug response . (**A**) Schematic representation of the CRISPR/Cas9-induced homologous recombination strategy to introduce the KRAS[G12D] mutation in the KRAS[WT] patient-derived CRC organoid P18T. Green bar: start codon. Red bar: G12D mutation. Parental and mutant sequences are shown on the right. (**B**) Extensive dual-inhibitor dose-response assay of patient-derived CRC organoids P18T and P18T-KRAS[G12D] treated for 72 hr. 14×14 drug concentrations of afatinib and selumetinib were chosen with

*Figure 3 continued on next page*

*Figure 3 continued*

logarithmic intervals covering a 5 nM–5 µM range. The results of the full matrix screen are represented as a heat map (left), where red represents 0% ATP levels (no viability) and green represents 100% ATP levels (max viability). The dose-response curves to the right represent the horizontal (afatinib monotherapy), vertical (selumetinib monotherapy) and diagonal (afatinib/selumetinib combination therapy) lines in the heat maps. Dashed lines are P18T; solid lines are P18T-KRAS$^{G12D}$. (C) Stills from representative time-lapse imaging (three days) of CRC organoids P18T and P18T-KRAS$^{G12D}$ treated with vehicle (DMSO) or afatinib + selumetinib (both 1 µM) (see also *Video 1*). In every panel, upper images show color-coded depth of maximum-projected z-stacks of H2B-mNeonGreen fluorescent organoids. Lower panels: corresponding transmitted light images. Time interval: 15 min. Scale bars: 20 µm. Representative time-lapse of 2 (total eight organoids/condition) and four experiments (total 20 organoids/condition) for P18T and P18T-KRAS$^{G12D}$ resp. (D) Mitotic and apoptotic events in the organoid drug response movies (C and *Video 1*) were manually marked and quantified (see Materials and methods and *Figure 3—figure supplement 3*). In comparison with vehicle (-), drug treatment of p18T with afatinib and selumetinib (a+s) results in both proliferation block and apoptosis induction, while p18T-KRAS$^{G12D}$ only shows reduced proliferation but unchanged apoptosis rates. Error bars represent standard deviation. *p<0,05; ***p<0,001; n.s. = not significant (p=0,4)

The following figure supplements are available for figure 3:

**Figure supplement 1.** Original heat map of viability and heat map of calculated scores for p18T and p18T-KRAS$^{G12D}$.

**Figure supplement 2.** Quantifying life and death during real-time imaging of drug response.

**Figure supplement 3.** Drug response of CRC organoids as examined by Western blot.

**Figure supplement 4.** Stills from representative time-lapse imaging (three days) of CRC organoids P18T and P18T-KRAS$^{G12D}$ treated with vehicle (DMSO) or a combination of targeted inhibitors afatinib (33 nM) and selumetinib (200 nM) (see also *Video 2*).

AKT or mTOR inhibitors in KRAS mutant CRCs did not yield satisfactory results (*Shimizu et al., 2012*).

## Response profiles to targeted inhibition of the EGFR-RAS-ERK pathway are comparable in normal and tumorigenic organoids

Next, we aimed to further establish whether the effects of oncogenic KRAS on drug response is dependent on a tumorigenic background or could manifest independent of cellular state. We therefore used normal colon organoids and a derivative of that line in which the oncogenic *KRAS$^{G12D}$* mutation was introduced via similar CRISPR/Cas9-mediated genome-editing strategy as in P18T (*Drost et al., 2015*). In analogy with mouse studies (*Snippert et al., 2014*), we observed no morphological alteration nor induction of senescence upon introduction of oncogenic KRAS (*Figure 5—figure supplement 1*). Strikingly, drug response profiles of normal organoids to targeted inhibitors against the EGFR-RAS-ERK pathway (*Figure 5* and *Figure 5—figure supplement 2*) revealed a similar trend as in CRC organoid P18T (*Figure 4* and *Figure 4—figure supplement 1*). Thus, the effect that oncogenic KRAS imposes on drug response appears independent of cellular status and the presence of additional cancer mutations.

## Screening a panel of human CRC organoids confirms the differential effect of EGFR inhibition

Next, we aimed to extend our analyses towards a wider collection of CRC organoids that is more representative for the clinic. We screened 10 additional patient-derived CRC organoids for combinatorial therapies against the EGFR-RAS-ERK signaling pathway. Since all the organoid lines are fully characterized in terms of genome information, we could select CRC organoids with and without a mutant RAS pathway. Based on EGFR/HER2 dual inhibition by afatinib, we could clearly discriminate the organoid panel in two types of responders, namely the sensitive versus the resistant ones (*Figure 6A*, green versus red lines respectively). Indeed, drug sensitivity towards all tested EGFR inhibitors clearly correlated with the mutational status of *KRAS*. However, there were two notable exceptions (*Figure 6A* and *Figure 6—figure supplement 1*). The first was P25T, which, although WT for *KRAS*, turned out to contain an oncogenic mutation in *NRAS* (Q61H), thereby fully explaining the resistant behavior. The second exception was organoid line P19bT that, unlike the other CRC organoids in our panel, is characterized as microsatellite instable (MSI) including the hyper-mutator

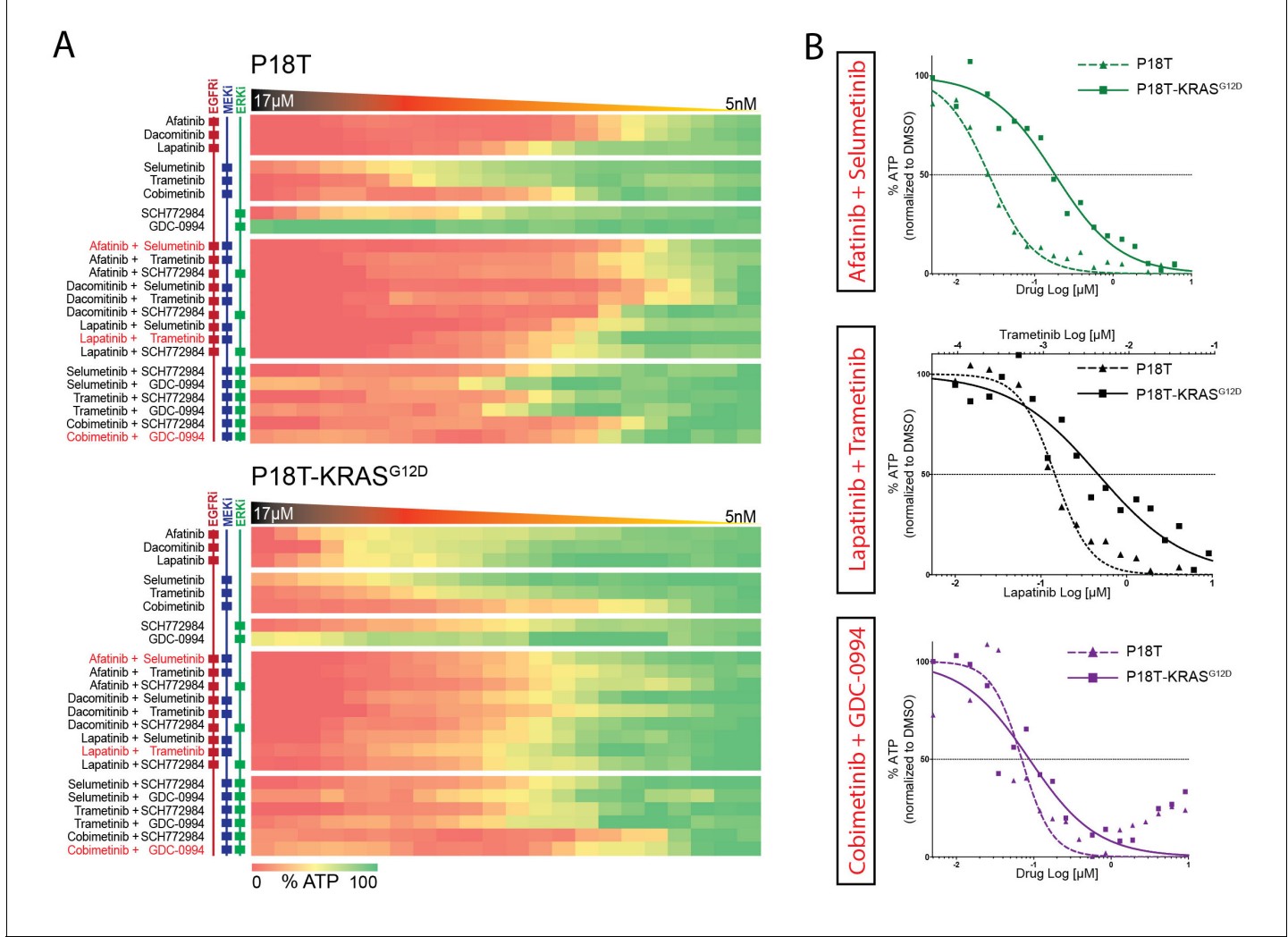

**Figure 4.** Differential drug sensitivities upon combination therapies including EGFR inhibition. (A) Heat map of dose-response measurements (cell viability) in CRC organoids P18T (top panel) and P18T-KRAS[G12D] (bottom panel). Organoids were treated (72 hr) with vehicle (DMSO) or inhibitors targeting the EGFR-RAS-ERK pathway (5 nM – 20 μM range, in 22 logarithmic intervals). Red represents 0% ATP levels (max cell death) and green represents 100% ATP levels (max viability). Drug names and their nominal targets are indicated in the left panel. Combination therapies that are currently in clinical trial for patients with RAS mutant CRCs are indicated in red font. See *Figure 4—source data 1* and *Supplementary file 1* for all dose-response curves. (B) Dose-response curves of CRC organoids P18T (dashed lines) and P18T-KRAS[G12D] (solid lines) treated with combination therapies that are currently in clinical trial for patients with RAS mutant CRCs.

The following source data and figure supplements are available for figure 4:

**Source data 1.** Dose-response curves for patient-derived tumor organoids P18T and P18T KRAS[G12D] as indicated.

**Source data 2.** Dose-response curves for patient-derived tumor organoids P18T and P18T KRAS[G12D] as indicated.

**Figure supplement 1.** Upper panel: Heat map of all IC50 values (Log10-scale) for P18T and P18T-KRAS[G12D], determined from dataset shown in *Figure 4A* (and *Figure 4—source data 1* and *Supplementary file 1*).

**Figure supplement 2.** Drug combinations on P18T and P18T-KRAS[G12D] organoids targeting EGFR-RAS-ERK and PI3K-AKT pathways.

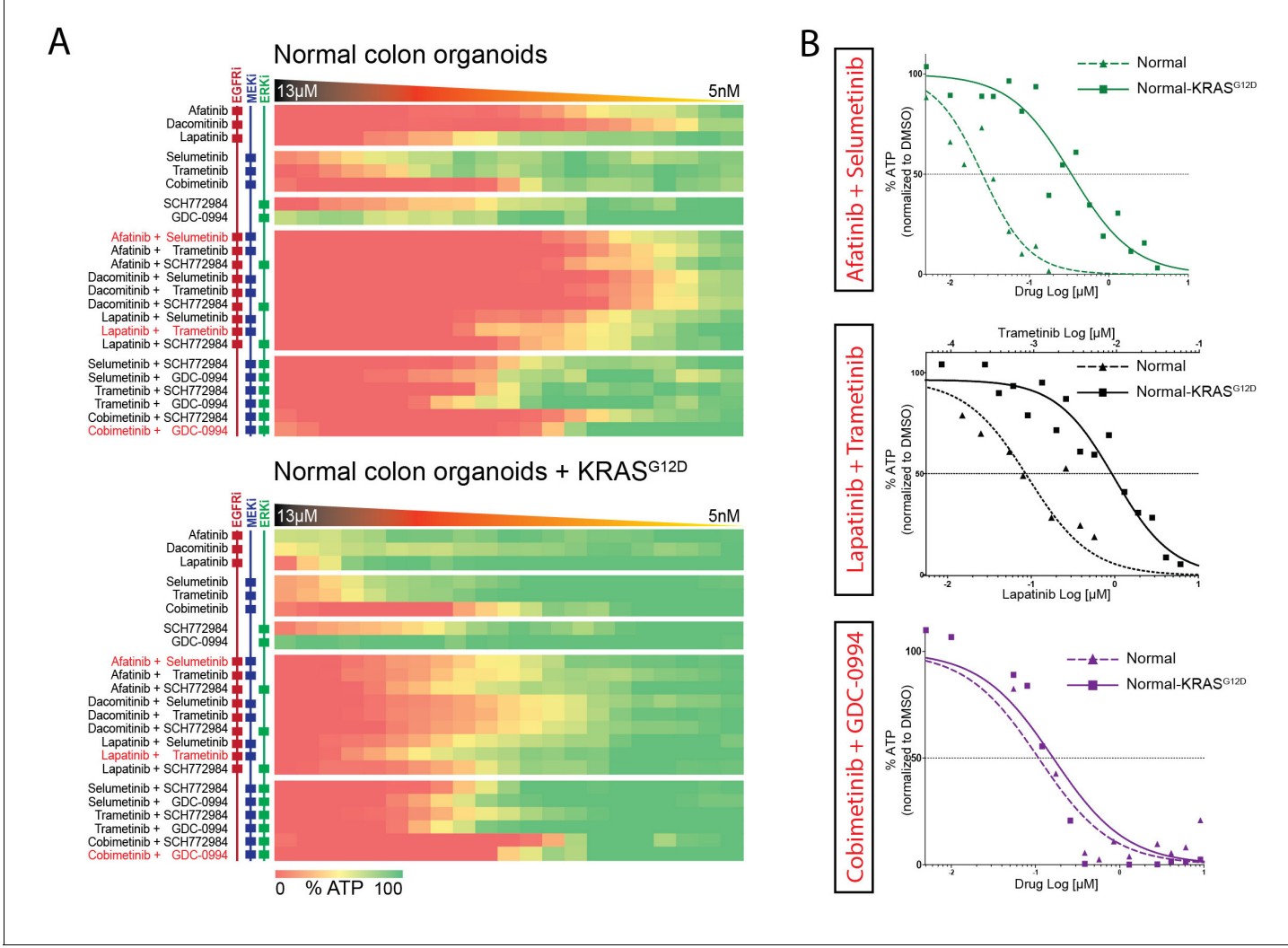

**Figure 5.** Comparable drug response profiles in normal and tumorigenic background. (**A**) Heat map of dose-response measurements of cell viability in normal colon organoids (top panel) and in normal colon organoids with an oncogenic KRAS mutation (bottom panel) after 72 hr drug treatment with inhibitors targeting the EGFR-RAS-ERK pathway. Same concentration range and color-coding as in *Figure 4*. Combination therapies that are currently in clinical trial for patients with RAS mutant CRCs are indicated in red. See *Figure 5—source data 1* and *Supplementary file 1* for all dose-response curves. (**B**) Dose-response curves of normal organoids (dashed lines) and normal organoids + KRAS (solid lines) treated with combination therapies that are currently in clinical trial for patients with RAS mutant CRCs.

The following source data and figure supplements are available for figure 5:

**Source data 1.** Dose-response curves for normal and normal KRAS$^{G12D}$ organoids as indicated.

**Figure supplement 1.** Comparison of normal organoids and normal organoids with an introduced oncogenic G12D mutation within the endogenous *KRAS* locus.

**Figure supplement 2.** Upper panel: Heat map of all IC50 values (Log10-scale) for normal colon organoids with and without mutant KRAS, determined from dataset shown in *Figure 5A* (and *Figure 5—source data 1* and *Supplementary file 1*).

phenotype (*van de Wetering et al., 2015*). Most importantly, P19bT tumor contains a *BRAF* (V600E) mutation, providing resistance towards the targeted drugs (*Di Nicolantonio et al., 2008*). Thus, these two cases underscore that drug screening on human organoids is able to evaluate the functionality of entire oncogenic pathways beyond the scope of the most frequent candidate mutations.

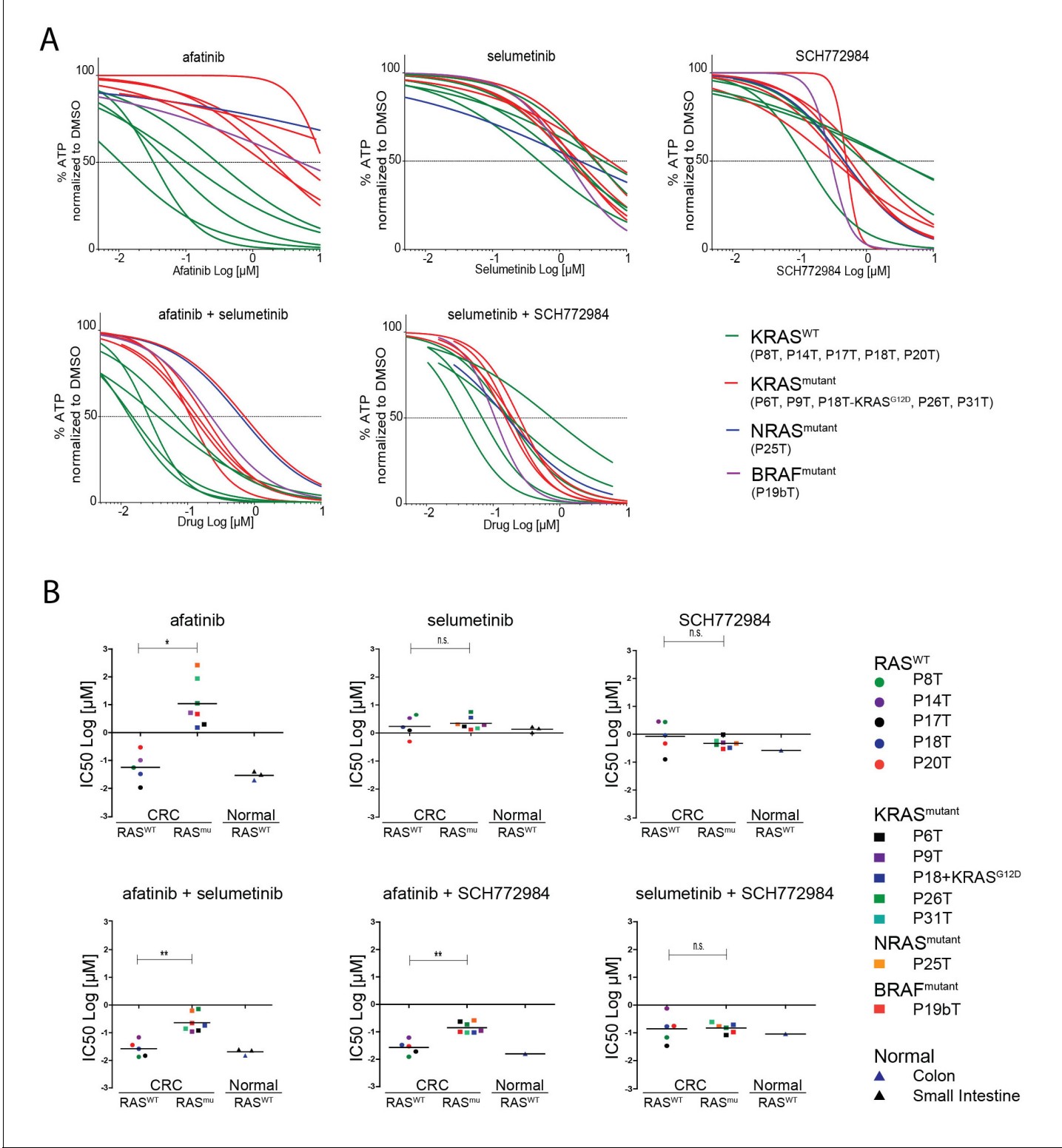

**Figure 6.** Screening multiple human CRC organoids confirm RAS mutational status for outcome EGFR inhibition. (**A**) Dose-response curves of 11 different patient-derived CRC organoids and one engineered CRC organoid (P18T-KRAS$^{G12D}$) treated for 72 hr with single targeted inhibitors or combinations thereof, namely afatinib (dual EGFR/HER2 inhibitor), selumetinib (MEK inhibitor) and SCH772984 (ERK inhibitor). 5 CRC organoids contain WT KRAS (green lines), 5 CRC organoids contained annotated oncogenic KRAS mutations (red lines), P19bT contains an oncogenic version of BRAF and P25T contains an oncogenic mutation in NRAS (purple and blue lines, resp.). See *Figure 6—source data 1* and *Supplementary file 1* for all dose-response curves. (**B**) CRC and normal organoids classified based on the mutational status of the RAS-RAF-MEK-ERK signaling pathway. Responses to

*Figure 6 continued on next page*

*Figure 6 continued*

afatinib, selumetinib, SCH772984 and combinations thereof, are shown in scatter plots of IC50 values (µM; 10log scale). Each colored dot represents an individual organoid line. Note that the experiment included normal organoids from the colon as well as the small intestine (three individual persons). Color corresponds as indicated in the legend. Black bar is the geometric mean. n.s., not significant. *p<0,05. **p<0,01.

The following source data and figure supplement are available for figure 6:

**Source data 1.** Dose-response curves for panel of patient-derived tumor organoids as indicated.

**Figure supplement 1.** Top panel: Heat map of all IC50 values (Log10-scale) for multiple drug responses in CRC organoids with and without mutant RAS signaling, determined from datasets shown in *Figure 6—source data 1* and *Supplementary file 1*.

Next, we aimed to evaluate the effects of drug combinations on CRCs organoids with WT and mutant RAS pathways, as well as on non-tumorigenic normal organoids. In all seven independent CRC patients with a mutant RAS pathway, we observed synergistic activity when combining EGFRi and MEKi (*Sun et al., 2014*) or EGFRi and ERKi (*Figure 6B*). Nevertheless, RAS WT organoids, either of CRC or normal origin, reveal higher sensitivities to combination therapies that include EGFR inhibition than RAS mutant CRCs. In contrast, therapies that do not include EGFR inhibition, but target MEK (selumetinib) and ERK (SCH772984), were similarly effective in all organoid lines, regardless of the mutational status of the RAS pathway or cellular state (*Figure 6B* and *Figure 6—figure supplement 1*).

Another important observation after comparative analysis on this wide CRC panel concerns patient-to-patient variability in the response to anti-EGFR monotherapy, even within the RAS WT and mutant subgroups (*Figure 6B*, upper left). Combining EGFRi with either MEK or ERKi results in a more consistent inhibitory effect over this set of CRCs (*Figure 6B*, lower panels), thereby not only improving individual responses but also augmenting success rates on a population scale. These findings are supportive to the previously published concept that proposes to combine EGFR with MEK inhibition directly at the start of therapy in patients with WT RAS tumors with the rationale of preventing sub-clones with acquired resistance to anti-EGFR monotherapy from reigniting tumor growth (*Misale et al., 2015*).

## Therapy-surviving cancer cells quickly restart tumor growth after release from targeted inhibition

Besides the direct effects of therapeutic treatments on tumor mass, the ability of cancer cells to recover from the treatments and restart tumor growth is of utmost relevance. We therefore studied the recovery of CRC organoids after release from (i.e. washout of) targeted inhibitors. More precisely, we monitored cellular viability, proliferation and cell death induction by quantifying viable nuclei (marked green) and dead nuclei (marked red) in 3D confocal tiled-scans at multiple time-points before and after treatment of targeted inhibitors (*Figure 7A* and Materials and methods).

In P18T, MEK or ERK inhibition (selumetinib or SCH772984 resp.) did not induce significant cell death after three days of monotherapy. Afatinib (EGFR/HER2i), either alone or in combination with selumetinib (MEKi), induced significant degrees of cell death, in line with our 72 hr monitoring of drug response. Similar effects were obtained by combined inhibition of MEK and ERK (*Figure 7B*, left panel). Additionally, this experiment shows that the described effects persisted for seven days after drug washout. As expected for P18T-KRAS$^{G12D}$, suppressing EGFR/HER2 activity upstream of mutant KRAS using afatinib proved ineffective, while monotherapy of MEKi or ERKi inhibited proliferation only to a minor extent (*Figure 7B*, right panel). Only the inhibitor combinations EGFRi/MEKi and MEKi/ERKi induced complete proliferative stagnation, albeit with minor induction of cell death. Importantly, independent of a therapeutic strategy, the CRC organoids quickly restored proliferative activity after drug release. Comparable results were obtained in CRC organoids P8T (KRAS WT) and KRAS mutant P26T, with the exception that cell death induction in P8T was less pronounced than in P18T (*Figure 7C*). In summary, these data indicate that inhibition of the EGFR-RAS-ERK pathway, independent of inhibitor combination, predominantly inhibits cell-cycle progression in KRAS mutant CRC organoids.

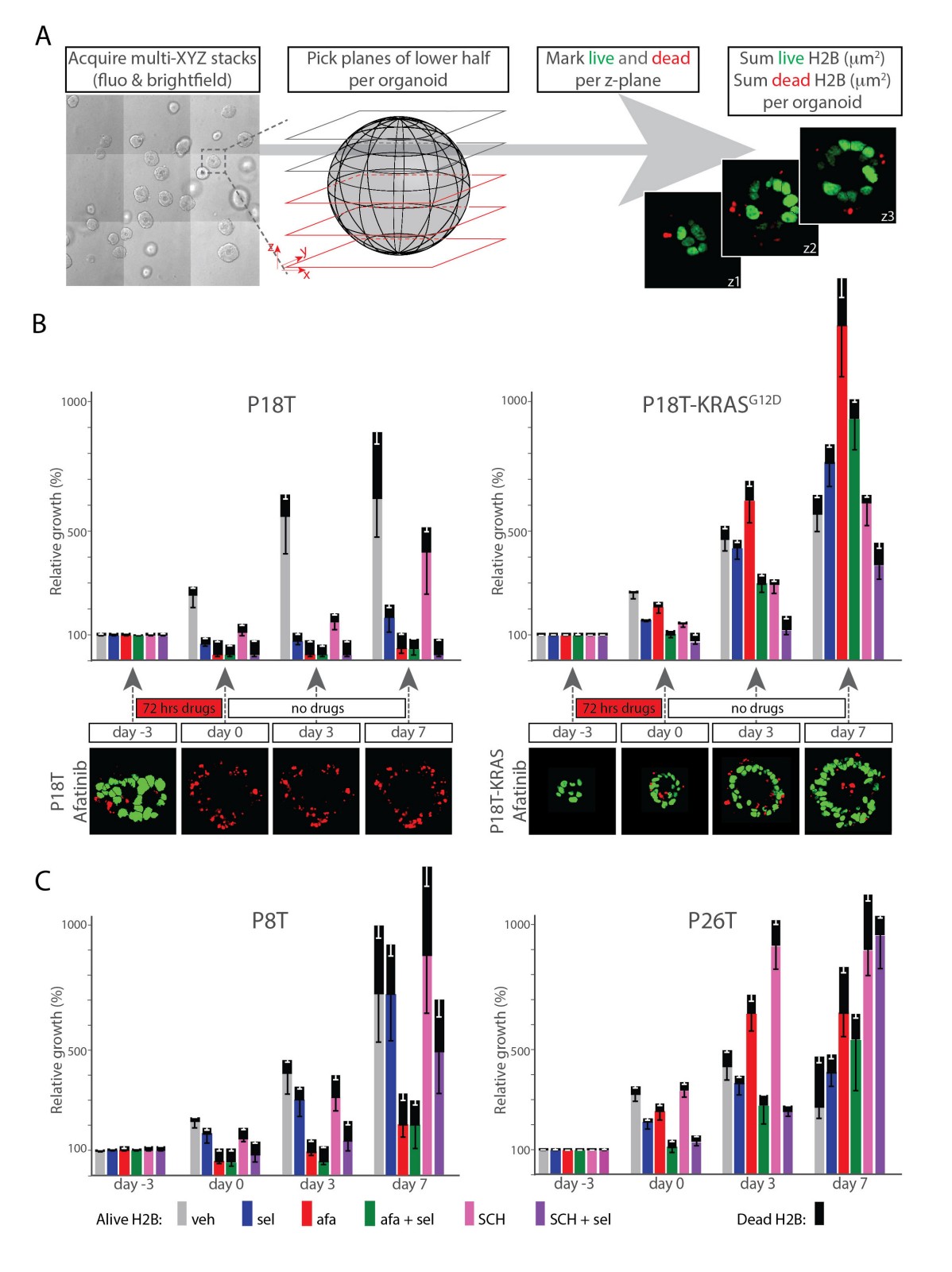

**Figure 7.** Therapy surviving cancer cells reignite proliferation after release of targeted inhibition. (**A**) Scheme of image-processing workflow. Multiple z-stacks were acquired in a tile-scan mode. H2B-mNeonGreen and bright field images were recorded of >10 organoids (left panel) over multiple days. Lower half of the imaged z-planes were selected of 3D-organoids that were fully recorded on each day (second panel). Live nuclei and dead nuclear remnants were marked for each z-plane, as identified by nuclear size (third panel, see Materials and methods section), measured and integrated per

*Figure 7 continued on next page*

*Figure 7 continued*

lower half of the 3D scanned organoid as an absolute measure for the amount of viable cells, while summed dead nuclei represent the amount of dead cells (fourth panel). (B) Bar diagrams showing proliferation and/or death of organoid cells following drug treatment and during recovery after drug removal. 3D tile-scans were acquired at the beginning and end of the therapy (day −3 and 0), as well as 3 and 7 days after the end of the therapy (i.e. drug removal) (day 3 and 7). All bars report pixel count from H2B-NeonGreen in living (color) as well as dead (black) organoid cells. Color corresponds to targeted inhibitor (see legend). All values are means ± s.e.m. of 12–15 organoids, normalized to 'alive H2B' prior to treatment. One representative z-plane is provided of a P18T and P18T-KRAS$^{G12D}$ CRC organoid during and after afatinib (dual EGFR/HER2 inhibitor) therapy. Green, alive nuclei. Red, marked nuclear remnants of dead cells. Color code legend is provided at the bottom of panel C. (C) Patient-derived CRC organoids P8T and P26T were treated and analyzed as described in B. In general, cancer cells that survived drug therapy rapidly reignited cell proliferation after drug release. veh, vehicle (DMSO); sel, selumetinib; afa, afatinib; afa+sel, afatinib+ selumetinib; SCH, SCH772984; SCH+sel, SCH772984+selumetinib.

The following source data and figure supplement are available for figure 7:

**Source data 1.** ImageJ/Fiji macro script: 'Macro Drug&release experiment'.

**Figure supplement 1.** The custom-made image analysis software for quantifications in *Figure 7* is extensively described in the Materials and methods Section.

## Dual inhibition of the EGFR-MEK-ERK pathway induces a G1 cell cycle arrest

To characterize the induced cell cycle arrest in RAS mutant tumor cells in more detail, we performed cell cycle analysis by flow cytometry using a 3 hr EdU pulse in combination with DNA staining to discriminate between the different cell cycle phases (G1, S and G2 respectively) in P18T-KRAS$^{G12D}$ and P26T. Indicative of a G1 arrest, we observed a sharp decline in the amount of RAS mutant tumor cells in S-phase using inhibitor combinations EGFRi/MEKi and MEKi/ERKi, while a similar fraction of cells accumulated in G1 (*Figure 8A*).

To further characterize drug-induced arrest, we investigated whether the regrowth after drugs washout involves all therapy-surviving tumor cells or only a minor subpopulation that fuels tumor relapse. For this, we performed two functional assays.

First, we performed EdU incorporations at various time points during the drug response and during organoid recovery after drug withdrawal. In agreement with the previous cell cycle analysis, almost no EdU incorporation was detected in the presence of inhibitor combinations EGFRi/MEKi and MEKi/ERKi (no cells in S-phase) (*Figure 8B*). However, during the first three days after drug withdrawal, the vast majority of growth-arrested tumor cells incorporated EdU again, suggesting renewed proliferative activity in virtually all tumor cells, thereby excluding the presence of senescence or minor subpopulations being responsible for restored growth (*Figure 8C*).

In addition, we performed live-cell imaging on tumor organoids after drug withdrawal and quantified the number of mitotic and apoptotic events over time (see Materials and methods). Indeed, in line with a G1 arrest, we observed a delay of about 20–24 hr after withdrawal of the drugs (EGFRi/MEKi) before observing numerous mitotic events again in all regions of the arrested organoids (*Figure 8D* and *Figure 8—figure supplement 1* and *Video 3*). (Similar results were obtained for MEKi/ERKi, data not shown).

## Robust inhibition of the EGFR-RAS-ERK pathway sensitizes for induced cell death

As described above, none of the therapies targeting the EGFR-RAS-ERK pathway could induce the desired degree of cell death in RAS mutant CRC organoids. As opposed to this, it has been reported that combined inhibition of anti-apoptotic BCL2 family members and effectors of the RAS pathway can effectively induce cell death in KRAS mutant cancers (*Corcoran et al., 2013*; *Hata et al., 2014*, *2015*; *Tan et al., 2013*). These results prompted clinical trials to evaluate combined targeting of MEK and BCL2/BCLXL in KRAS mutant solid tumors (NCT02079740). Unfortunately, clinical application of BCL2/BCLXL inhibitors is hindered by on-target toxicity of BCLXL inhibition in blood platelets (*Hata et al., 2015*) and might therefore strongly benefit from strategies that allow minimized doses.

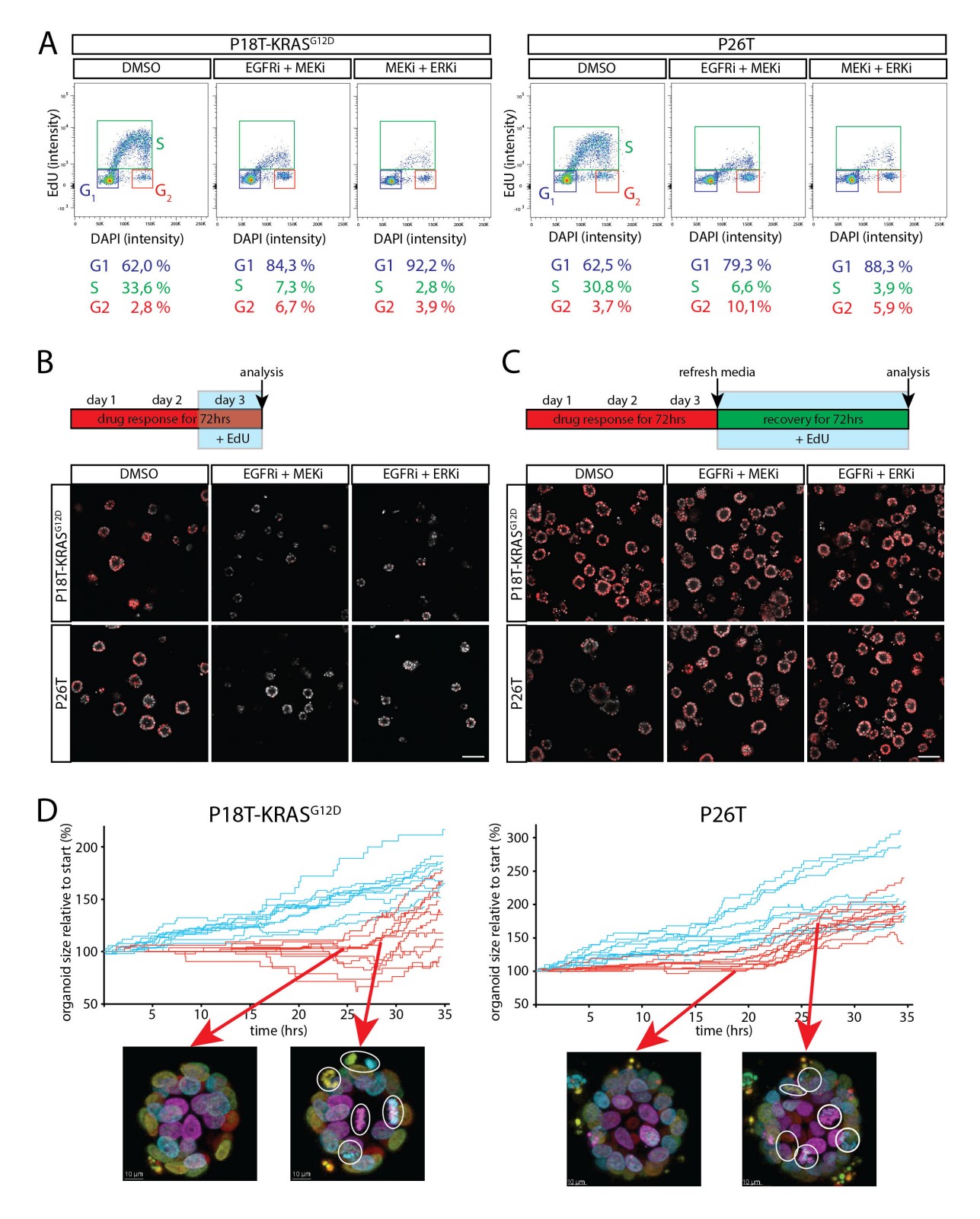

**Figure 8.** Cell cycle arrest upon dual inhibition of EGFR-MEK-ERK pathway. (**A**) Representative cell cycle analysis of P18T-KRASG12D and P26T by flow cytometry (n = 2). DNA was stained with DAPI and DNA-synthesis was detected using a 3 hr EdU pulse to clearly discriminate between G1, S and G2 stages of the cell cycle. Dual inhibition of the EGFR-MEK-ERK pathway significantly changes the distribution of cells between stages of the cell cycle (Chi2: all p values<0,0001) with a predominant increase in G1 at the expense of cells in S-phase. EGFRi + MEKi = afatinib + selumetinib. MEKi + ERKi =

*Figure 8 continued on next page*

*Figure 8 continued*

selumetinib + SCH772984. (B) Almost no incorporation of EdU (red) is detected during the last 24 hr of drug treatment using dual inhibition of the EGFR-MEK-ERK signaling pathway, indicative of halted proliferative activity. Nuclei are counterstained with Hoechst (white). EGFRi + MEKi = afatinib + selumetinib. EGFRi + ERKi = afatinib + SCH772984. Scale bar is 100 µm. (C) Virtually all cancer cells incorporate EdU (red) when provided after release from targeted inhibition of the EGFR-MEK-ERK pathway. Nuclei are counterstained with Hoechst (white). EGFRi + MEKi = afatinib + selumetinib. EGFRi + ERKi = afatinib + SCH772984. Scale bar is 100 µm. (D) Chronological ranking of mitotic and apoptotic events extracted from live-cell imaging data of tumor recovery reconstructs the organoid size evolution over time. In contrast to vehicle treated organoids (blue lines), afatinib + selumetinib treated organoids (red lines) show first mitotic activity again from 20–24 hr onwards after drug withdrawal. Typical snapshots of live-cell imaging data are provided. White circles indicate mitotic events. Arrows indicate the organoid and moment of snapshot.

The following source data and figure supplement are available for figure 8:

**Source data 1.** ImageJ/Fiji macro script: 'Score Events macro'.
**Figure supplement 1.** Average growth speeds of the organoids were determined by linear fitting of the traces shown in *Figure 8D*.

Here, we explored the use of navitoclax, a clinically tested BCL2/BCLXL inhibitor, in targeted therapies against RAS mutant CRC organoids. Indeed, straightforward ATP-based screening confirmed that navitoclax, when combined with afatinib (*Figure 9A*, left panel), selumetinib (*Figure 9A*, middle panel), or both (*Figure 9A*, right panel) is far more efficient in antagonizing tumor organoid growth than any of the related monotherapies. Importantly, regarding the dose-limiting effects of navitoclax, we show that robust dual inhibition of the EGFR-RAS-ERK pathway (1 µM afatinib/1 µM selumetinib) is exceptionally potent in sensitizing navitoclax-induced effects (*Figure 9A*, right panel). Such strong sensitization could not be achieved by afatinib (1 µM) alone or selumetinib (1 µM) alone. Similar results were obtained in other patient-derived CRC organoids that harbor a *KRAS* mutation (*Figure 9—figure supplement 1*) or, alternatively, with different combinations of inhibitors targeting the EGFR-RAS-ERK pathway (*Figure 9—figure supplement 2*).

To ensure that the observed effects represent cell death rather than cell-cycle arrest, we performed qualitative and quantitative microscopic analyses (*Figure 9B*). Indeed, cell death could be induced by low concentrations of navitoclax when combined with high concentrations of afatinib and selumetinib.

Next, we designed a medium-throughput assay to monitor the persistence of drug response after wash-out of the above-mentioned inhibitor combinations (*Figure 9C*). As expected, even dual inhibition with afatinib (EGFR/HER2i) and selumetinib (MEKi) at high concentrations does not induce sufficient cell death, since these cultures can recover to the levels of untreated controls within six days (*Figure 9C*). In contrast, the addition of low concentrations navitoclax to high concentrations of inhibitor combination afatinib/selumetinib potently induces cell death, as shown by the sustained inhibitory effects on culture growth (*Figure 9C*, black bars).

In light of the dose-limiting toxicity of navitoclax in blood platelets, we performed full matrix-screens to explore optimal combinations of drug concentrations (*Figure 9D*). In agreement with previous results, the more efficient inhibition of the RAS pathway, i.e. high concentrations and/or dual targeting (*Figure 9—figure supplement 3*) the lower the concentration of navitoclax that is necessary to affect cellular viability. Furthermore, venetoclax, a BCL2-specific inhibitor, is unable to reproduce the effects of navitoclax (BCL2/BCLXLi) (*Figure 9D*), suggesting that in agreement with the reported findings in lung cancer

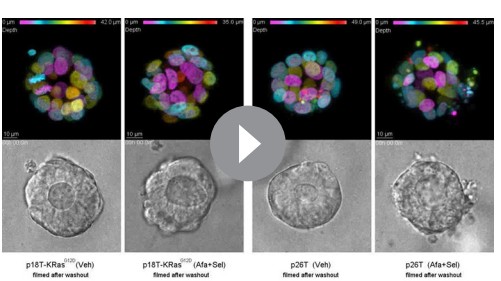

**Video 3.** Real-time imaging of cellular behavior in tumor organoids surviving treatment with afatinib and selumetinib. P18T-KRAS$^{G12D}$ and P26T organoids were treated for 72 hr with afatinib (1 µM) and selumetinib (1 µM), similarly to *Figures 1* and *3* and *Video 1*. After the subsequent washout of the drugs, organoids were imaged for ~40 hr to visualize cell behaviour in surviving organoids.

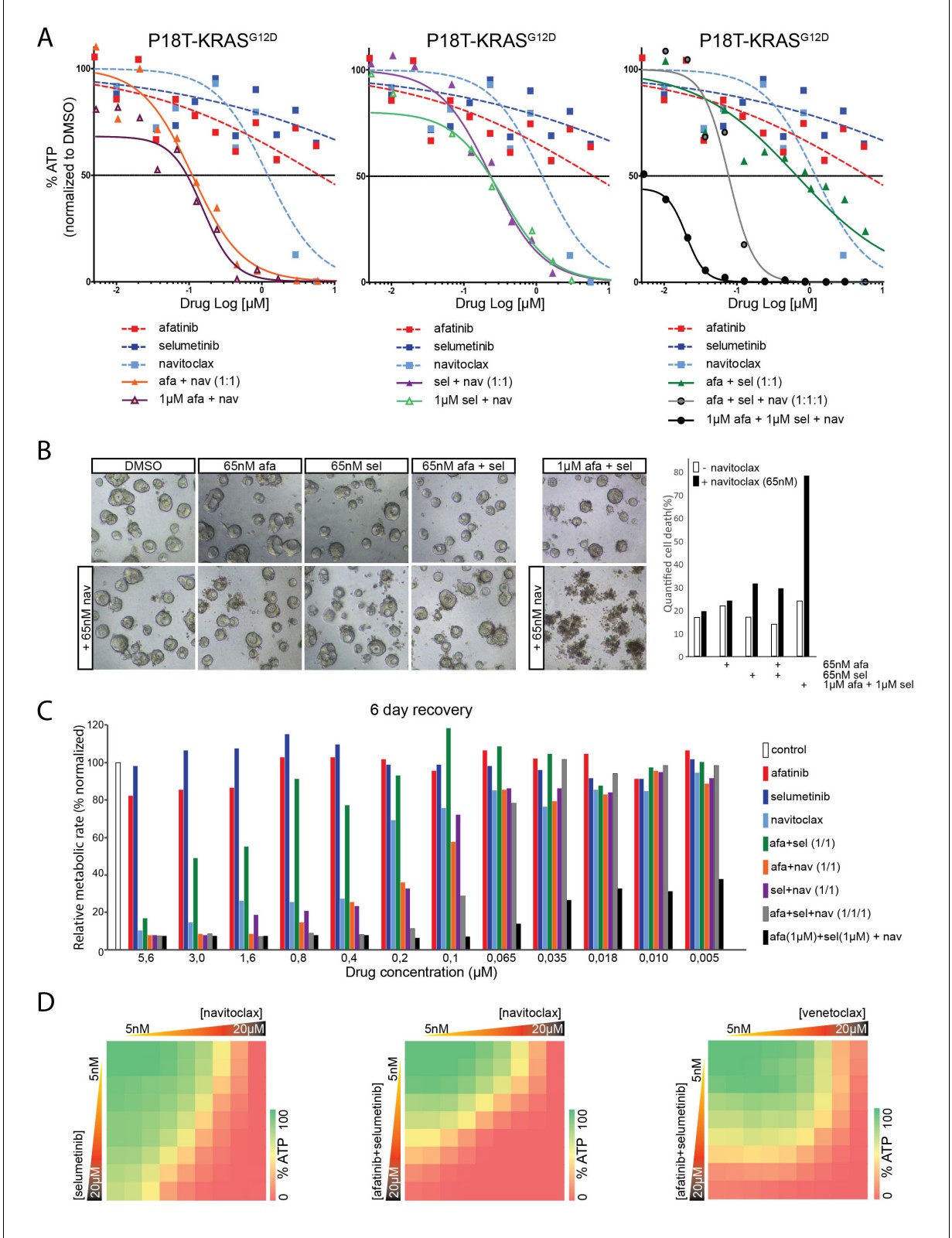

**Figure 9.** Robust inhibition of the EGFR-RAS-ERK pathway sensitizes for navitoclax-induced cell death. (**A**) Dose-response curves of patient-derived CRC organoids P18T-KRAS[G12D] treated with the dual EGFR/HER2 inhibitor afatinib, MEK inhibitor selumetinib, BCL2/BCLXL inhibitor navitoclax or a combination thereof. Cell viability was measured by an ATP-based assay after 72 hr of drug treatment. Inhibition of the EGFR-RAS-ERK pathway using high concentrations of afatinib and selumetinib (1 µM) strongly sensitizes for navitoclax-induced reduction of cellular viability (right panel, black line). *Figure 9 continued on next page*

*Figure 9 continued*

Such strong sensitization could not be achieved by afatinib (1 µM) alone (left panel, orange line) or selumetinib (1 µM) alone (middle panel, purple line). Dose-response curves are averages of n = 2. (B) Representative images taken from CRC organoids P18T-KRAS$^{G12D}$ treated for 72 hr with above described drug combinations. Low amounts of navitoclax (65 nM) only induce cell death in combination with effective inhibition of the RAS pathway using a high concentrations of afatinib and selumetinib. Bar diagram at the right shows quantifications of cell death by scoring propidium-iodide stained nuclei (dead) over viable H2B-labeled nuclei (see Materials and methods section) of minimal 15 organoids per condition (signals pooled prior ratioing, hence no standard deviation calculated: see Materials and methods section). Representative experiment of n = 2. (C) Bar diagrams representing cellular viability (alamarBlue assay)of organoids that have been recovered for six days after 72 hr drug treatment at different concentrations ranging from 5 µM (left) to 5 nM (right). Color corresponds to targeted inhibitor as indicated in the legend. All values are normalized to control samples (DMSO). Targeting of the RAS pathway does not provide long-lasting effects after drug removal, even combination treatments at high concentrations (green bars). However, it does sensitize for cell death induction using low amounts of navitoclax (black bars). An average of two independent experiments is shown. Bar diagrams are averages of n = 2. (D) Extensive dual or triple-inhibitor dose-response assay of patient-derived CRC organoids P18T-KRAS$^{G12D}$ treated for 72 hr. 9×9 drug concentrations of selumetinib (MEKi) or afatinib/selumetinib (1/1) versus navitoclax (BCL2/BCLXL) or venetoclax (BCL2) were chosen with logarithmic interval covering a 5 nM–20 µM range. The results of the full matrix screen are represented as heat maps, where red represents 0% ATP levels (no viability) and green represents 100% ATP levels (max viability). Exploring optimal drug concentrations reveal that the more effective inhibition of the RAS pathway is achieved (dual targeting and high concentrations), the less navitoclax is required. Venotoclax, a BCL2-specific inhibitor, is not able to copy the effects of navitoclax (BCL2/BCLXL).

The following source data and figure supplements are available for figure 9:

**Source data 1.** ImageJ/Fiji macro script: 'Macro PI versus H2B'.

**Figure supplement 1.** Dose-response curves of patient-derived KRAS mutant CRC organoids P9T and P26T treated with the dual EGFR/HER2 inhibitor afatinib, MEK inhibitor selumetinib, BCL2/BCLXL inhibitor navitoclax or a combination thereof.

**Figure supplement 2.** Dose-response curves of patient-derived CRC organoids P18T-KRAS treated with different combination therapies against the EGFR-RAS-ERK pathway with addition of BCL2/BCLXL inhibitor navitoclax and the corresponding mono and dual therapy controls.

**Figure supplement 3.** Drug response of P18T-KRAS$^{G12D}$ and P26T CRC organoids examined by Western bot after 24 hr.

(*Corcoran et al., 2013*), it is BCLXL that protects against apoptosis upon targeted inhibition of a mutant RAS pathway in CRC organoids (*Figure 9D*).

## Discussion

Patient-derived CRC organoids were recently introduced as a model system in cancer research that is complementary to cell lines and PDX models (*van de Wetering et al., 2015*). We assembled a panel of normal and CRC organoids with either WT or mutant RAS that had been derived from different patients (*van de Wetering et al., 2015*). Moreover, we included normal (*Drost et al., 2015*) and tumor organoids in which the oncogenic G12D mutation was introduced in the endogenous *KRAS* locus by CRISPR-Cas9-induced homologous recombination. These engineered organoid lines, in combination with patient-derived CRC organoids of different genetic backgrounds, allowed us to study the effect of mutant KRAS on drug response to targeted inhibition of the EGFR-RAS-ERK pathway. Moreover, real-time imaging allows the monitoring of exact cellular fates in drug-challenged CRC organoids with spatial (3D) and temporal resolution.

Using this panel, we show that the presence of mutant RAS is sufficient to confer resistance to EGFR inhibitors. Moreover, we confirmed the synergistic effect of the clinically tested combination of pan-HER and MEK inhibition on mutant RAS organoids. However, we found that RAS mutant organoids remained largely resistant to apoptosis and became largely arrested in proliferation. Importantly, for KRAS mutant tumor organoid P26T we observed similar drug sensitivities in vitro as in vivo upon xenotransplantation. Moreover, similar growth arrested responses were reported in PDX mouse models of KRAS mutant CRC cancers ( *Sun et al., 2014*), underscoring the notion that tumor organoids are a reliable model system to test drug responses.

We report, for the first time, how normal tissue organoids respond to inhibitors targeting the EGFR-RAS-ERK pathway. Intriguingly, drug effects were almost identical in normal organoids and patient-derived CRC organoids when WT for RAS. This may imply that the sensitivity of RAS WT

colon cancer for EGFRi is not an acquired oncogene-addiction, but merely represents the dependency of normal colon (stem) cells on EGFR signaling activity (*Wong et al., 2012*). Indeed, normal organoids from the human colon consist predominantly of proliferative stem and progenitor cells due to the WNT ligands in the culture medium. Therefore, the observed toxicity in the normal organoids may very well be most representative for direct effects on the stem cell compartments of the normal colon. Indeed, one of the direct side effects of anti-EGFR monotherapy is diarrhea (*Miroddi et al., 2015*). In analogy to drug responses with WT RAS, normal organoids harboring a CRISPR-introduced oncogenic *KRAS* mutation showed resistance profiles towards targeted therapies that closely resemble those of RAS mutant CRC organoids, again underscoring the dominance of the RAS mutational status on drug response.

Anti-EGFR therapy in patients with KRAS WT colon tumors is standard of care, whereas patients with RAS mutant tumors are excluded. Our results confirm the drug sensitivity profile of colorectal cancers with and without mutant RAS, as has been established both in other model systems as well as in the clinic (*Sun et al., 2014*; *Karapetis et al., 2008*; *Amado et al., 2008*). For RAS mutant tumors, a number of different drugs and drug combinations have been tested, but thus far this has been without significant effect (*Ryan et al., 2015*). Our analyses confirm that drug treatments targeting the EGFR-RAS-ERK and the PI-3K/AKT cascades, including combinations thereof, are largely ineffective in RAS mutant CRC organoids.

In contrast, for CRC patients with RAS WT tumors combination therapies that target the EGFR-MEK-ERK pathway may be an improved alternative over anti-EGFR monotherapy. First, we observed that this combination treatment induced cell death more systematically over a panel of individual patient-derived CRC organoids with WT RAS (i.e. with decreased variability) than monotherapy with a pan-HER inhibitor. Furthermore, combination treatments may decrease the potency of low-abundant RAS mutant subclones to initiate tumor-relapse during therapy against a predominantly KRAS WT tumor (*Misale et al., 2015*).

The combined inhibition of pan-HER and MEK is currently tested in patients with RAS mutant cancers in several clinical trials (e.g. NCT02450656, NCT02230553 and NCT02039336). Also in our hands, this combination showed a clear synergistic effect in suppressing growth of RAS mutant CRC organoids, as determined by a straightforward ATP-based assay. Importantly however, our data revealed that this inhibitor combination induced a cell cycle arrest in mutant RAS organoids but no cell death. As a result, the cells rapidly restored proliferative activity after withdrawal of these drugs. The inability to induce cell death likely affects the long-term effectiveness of this combination in CRC patients with mutant RAS.

An alternative combination that is currently in clinical trials is the combined inhibition of MEK and ERK (NCT02457793). The rationale behind this combination is the notion that resistance to targeted inhibition of RAF and MEK often involves reactivation of ERK (*Ryan et al., 2015*), while suboptimal suppression of ERK activity in RAF mutant cancers may underlie the limited efficacy (*Bollag et al., 2010*; *Corcoran et al., 2015*). Although our RAS mutant CRC organoids showed sensitivity to dual inhibition of MEK and ERK, also this drug combination induced cell-cycle arrest rather than cell death, questioning whether it will be sufficient for the treatment of RAS mutant CRC.

With respect to clinical applications, we here report that effective inhibition of the EGFR-MEK-ERK pathway through combinatorial targeting does significantly prime the cytostatic RAS mutant cancer cells for apoptosis. This can be utilized by low concentrations of navitoclax, one of the most clinically advanced inhibitors of anti-apoptotic BCL family members. Minimizing navitoclax concentrations would be beneficial due to its on-target effects on BCLXL in circulating platelets, thereby causing thrombocytopenia (*Hata et al., 2015*). However, triple combination therapy with low concentrations of navitoclax (50 mg/kg; five days on, two days off) proved to be too toxic for the mice (data not shown). Nevertheless, we consider the navitoclax-induced apoptosis as a proof-of-principle that EGFR-MEK-ERK pathway inhibition in combination with alternative signaling nodes holds great promise in identifying therapeutic drug combinations that kill RAS mutant tumor cells while being tolerated by the patient.

In summary, we show drug responses of a wide panel of patient-derived CRC organoids to multiple clinically advanced targeted inhibitors, either alone or in combinations, against the EGFR-RAS-ERK pathway. Importantly, the drug phenotypes that we observe in the organoids appear representative for previous reported responses in vivo. We believe that organoid collections will facilitate the identification and optimization of effective targeted therapies, since drug screens can be performed

at a scale that is currently unprecedented when using resource-intensive PDX models. Due to the reliability and scalability of tumor organoids as a model system, we advocate that novel drugs should be tested on a panel of tumor organoids before their use in clinical trials.

## Materials and methods

### Patient-derived organoid culture and maintenance

The patient-derived organoids used in this study were previously established and characterized (*van de Wetering et al., 2015*). Human CRC and healthy colon organoids were cultured as described previously (*van de Wetering et al., 2015*). In short, organoids were cultured in drops of Basement Membrane Extract (BME; Amsbio) and medium was refreshed every two days. The CRC culture medium contained advanced DMEM/F12 (Invitrogen) with 1% Penicillin/Streptomycin (P/S, Lonza), 1% Hepes buffer (Invitrogen) and 1% Glutamax (Invitrogen), 20% R-spondin conditioned medium, 10% Noggin conditioned medium, 1X B27 (Invitrogen), 1.25 mM n-Acetyl Cysteine (Sigma-Aldrich), 10 mM Nicotinamide (Sigma-Aldrich), 50 ng/ml EGF (Invitrogen), 500 nM A83-01 (Tocris), 10 µM SB202190 (ApexBio) and 100 µg/ml Primocin (Invitrogen). The medium of healthy colon organoids had additional Wnt conditioned media. Organoids were splitted through shear stress (pipetting) and/or Trypsin-EDTA (Sigma-Aldrich) treatment. Culture medium after splitting was supplemented with 10 µM Y-27632 dihydrochloride. Organoid cultures have repeatedly been tested negative for Mycoplasma. Western blots are performed as described previously (*Drost et al., 2015*). Antibodies used: ERK (RRID:AB_390779), pERK (RRID:AB_331646) and GAPDH (RRID:AB_2107445).

### Vector construction, organoid transfection and genotyping

CRISPR guide RNAs (sgRNAs) were generated as described by *Drost et al. (2015)*. The KRAS target sequences used were: 5′-GAATATAAACTTGTGGTAGTTGG-3′ and 5′-GTAGTTGGAGCTGGTGGCG TAGG-3′. Transfections of p18T and p26N organoids with sgRNAs and subsequent selections by withdrawing EGF and adding the EGFR inhibitor gefitinib to the culture medium were performed as previously described (*Drost et al., 2015*). The presence of KRAS G12D mutation was verified by sequencing the PCR product obtained using the primers *KRAS*_for, 5′-TGGACCCTGACATAC TCCCA-3′ and *KRAS*_rev, 5′-AAGCGTCGATGGAGGAGTTT-3′ (*Drost et al., 2015*).

### Lentiviral transduction

Organoids were infected with lentivirus encoding histone2B fused to mNeonGreen (bright monomeric GFP variant) linked to a puromycin-resistance gene (pLV-H2B-mNeonGreen-ires-Puro) (*Shaner et al., 2013*) to visualize and track nuclei. Infected organoids were selected using 2 µg/ml puromycin.

### Drug screen and viability assay

Five days after organoid typsinization, 1 mg/ml dispase II (Invitrogen) was added to the medium of the organoids and these were incubated for 15 min at 37°C to digest the BME. Subsequently, organoids were mechanically dissociated by pipetting, filtrated using a 40 µm nylon cell strainer (Falcon), resuspended in 2% BME/growth medium (15–20,000 organoids/ml) prior plating of 30 µl (Multidrop™ Combi Reagent Dispenser) on BME pre-coated 384-well plates. The drugs and their combinations were added 3 hr after plating the organoids using the Tecan D300e Digital Dispenser. Drugs were dispensed in a randomized manner and DMSO end concentration was 0.4% in all wells. 72 hr after adding the drugs, ATP levels were measured using the Cell-Titer Glo2.0 (Promega BV) according to the manufacturer's instructions. Results were normalized to vehicle (DMSO = 100%) and baseline levels (multi drug ATP plateau at high concentrations = 0%) that were manually determined per organoid type and screening day. Multiple identical drug combinations were averaged. Heatmaps were smoothened using a moving average. Bliss scores were calculated as described previously (*Tan et al., 2013*).

### Targeted inhibitors

Afatinib, Dacomitinib, Lapatinib, Selumetinib, Trametinib, BYL719, MK2206 and GDC-0994, navitoclax and venetoclax were purchased from Selleck Chemicals. SCH772984 was obtained from

MedChem Express and Cobimetinib from Active Biochem. These compounds were dissolved in dimethylsulfoxide (DMSO, Sigma-Aldrich) and stored as 10 mM aliquots.

## Curve fitting of drug sensitivity

Data analyses were performed using GraphPad software by applying the nonlinear regression (curve fit) and the equation log(inhibitor) vs. normalized response (variable slope).

## Time-lapse imaging

Five days after trypsinization, H2B-mNeonGreen-expressing organoids were plated in matrigel in glass-bottom 96-well plates and mounted on an inverted confocal laser scanning microscope (Leica SP8X) under controlled conditions (37°C, 6% $CO_2$). Drugs were added to the organoids just prior to imaging. For 72 hr, organoids were imaged every 15 min in XYZT-mode using a 40x objective (1.1NA) and a 506 nm laser excitation light from a tunable white light laser for 72 hr. The images were converted using ImageJ/Fiji software into manageable and maximally informative videos, combining z-projection, depth color-coding and merging with transmitted light images (see source code files, 'Organoid Movie Macro').

## Drug and release assay by microscopy

Five days after trypsinization, H2B-mNeon-expressing organoids were plated in glass-bottom 96-well plates. Prior to drug addition (day −3), the organoids were imaged on a Leica SP8X. One 3D tile scan (merging 3×3 images, ~175 µm in Z in total, 5 µm Z-step) was acquired per well, allowing the visualization of 10–20 organoids per well. On day 0, 3 and 7, exactly the same fields of organoids were imaged again and medium was refreshed. A custom-made ImageJ/Fiji macro (see source code files, 'Macro Drug&Release') was developed to analyze 12–15 organoids per well/condition in a paired manner, i.e. individual organoids were tracked over the entire experiment (>10 days). Per organoid, a pseudo-quantitative measure for absolute numbers of living and dead cells was established as follows: (1) Thresholding on H2B-Neon fluorescence to select H2B-positive pixels (total). (2) Marking of 'dead' pixels in each slice (initially automated, based on particle size and eventually manually by selection). (3) Dividing pixels in dead and alive (total minus dead). (4) Integrating the pixel areas representing alive and dead H2B, respectively, from the slices that make up for the lowest 50% of the organoid volume. This was done to avoid analysis on the upper 50% of the volume, which is inevitably of lower image quality.

The analysis was performed in 12 to 15 organoids per well/condition. No biological replicates.

Of note, this method is independent of non-permeable DNA dyes such as PI to avoid their potential long-term effect on organoid growth. In order to validate the current method, a single time point data set was acquired with the use of PI (see *Figure 7—figure supplement 1*), validating the robustness of the strategy.

## Medium-throughput drug and release assay

For the analysis of organoid recovery after drug withdrawal, organoids were treated with the indicated drugs for 72 hr as described. Subsequently, drugs were washed out through aspiration of the drug-containing medium, followed by a washing step (10' incubation at 37°C with basal DMEM/F12 (+++)). After washing, organoids were incubated with CRC-medium containing 2%BME for recovery. After 48 hr, the medium was replaced by CRC-2%BME combined with 10% AlamarBlue (AB) cell viability reagent (ThermoFisher Scientific) according to the manufacturer's instructions. The increase of AB fluorescence (excitation 544 nm, emission 590 nm) was monitored over a time course of 2 hr (with measurements at 15' intervals) at 37°C, using a SpectraMax M5 microplate reader (Molecular Devices). Fluorescence kinetics were plotted over time (as relative fluorescent units (RFU) per hour) to define the linear range of the assay. Cell viability was then defined as the maximum AB fluorescence (RFUmax, within the linear range of the assay), corrected for background fluorescence (RFU at time point 0). Viability data from drug treated organoids were normalized to vehicle (DMSO) treated controls. Upon the AB time course, organoids were washed two times with basal DMEM/F12 and incubated with CRC-2%BME for another 96 hr, after which the AB assay was repeated.

### Quantification in H2B-Neon-expressing organoids using proidium iodide

H2B-mNeon-expressing organoids were plated in 384-well plates and provided with drugs as described for the viability assay. To selectively stain dead cells, propidium iodide (PI) was added 2 hr before starting imaging (2 μM). 3D stacks of 150 μm (7,5 μm per plane) were acquired on a Leica SP8 scanning confocal microscope, using a 10x dry lens for large field-of-view (3000×3000 pixels). Being an endpoint assay, high laser intensities could be applied for optimal imaging quality (and hence analysis). Green (mNeon) and red (PI) signals were sequentially acquired to avoid spectral mixing. Signals were pooled for 15–20 organoids before ratio calculation, hence no error bars.

Custom-made software (ImageJ/Fiji, see source code files, 'Macro PI versus H2B') was designed to determine surfaces (i.e. numbers of pixels) representing H2B-mNeon and PI, respectively; the ratio of these surfaces (PI/H2B) is the quantitative measure for cell death in the drug-challenged organoids. Central to the unambiguous determination of these surfaces is setting of the threshold. Our algorithm initially measures surfaces and corresponding mean intensities with ramping threshold, and from these data mathematically derives the threshold-optimum by combining the highest mean signal and most confined surface area.

### Quantification of mitotic and apoptotic events of live-cell imaging data

Depth-coded projection movies were analyzed for life and death in time: mitotic and apoptotic events were marked with help of custom-made ImageJ/Fiji macro (see source code files, 'ScoreEvents'). Indicated events were automatically drawn in the movie (essential when aiming to mark all events) and data were automatically sorted into Excel-files containing a (chronologically sorted) list of events.

### Cell cycle analysis by flow cytometry

3 hr prior trypsinization (TriplE, 5 min at 37C) to a single cell, organoids were incubated with 500 nM EdU. Single cells were fixed with ethanol (5%). EdU click-it reaction was performed according to manufacturer's protocol. DNA was stained using 1 μg/ml DAPI. Cells were analysed using a FACS-Canto II (BD).

### Organoid xenograft experiments

Approval for this study was obtained by the local animal experimental committee at The Netherlands Cancer Institute (DEC-NKI; OZP = 12012 and WP5727 and WP5689). P26T patient-derived organoids were trypsinized and 200,000 cells were resuspended in 50 μl medium/Matrigel (BD Biosciences) mixture at a 1:1 ratio and injected subcutaneously into NSG mice (JAX stock no: 005557). Mice with established tumors (average volume of 300 mm$^3$) were treated with afatinib (12.5 mg/kg; five days on, two days off), selumetinib (20 mg/kg; same schedule) or with a combination of both drugs for four weeks. After three weeks recovery from the drug treatment, mice were sacrificed.

For the second in vivo experiment, P26T organoids (~300.000 cells) were resuspended in 50% matrigel/medium with 10% collagen type I (BD Biosciences) and injected subcutaneously into NSG mice (JAX stock no: 005557). Mice with established tumors (average volume of 200 mm$^3$) were treated with afatinib (20 mg/kg; five days on, two days off), selumetinib (25 mg/kg; same schedule) or with a combination of both drugs for four weeks.

Tumor volumes were evaluated three times per week by caliper and the approximate volume of the mass was calculated using the formula $Dxd^2/2$, where D is the major tumor axis and d is the minor tumor axis. For in vivo dosing, afatinib was dissolved in 1.8% hydroxypropyl-b-cyclodextrin (Sigma), 5% of a 10% acetic acid stock and aqueous natrosol (0,5%). Selumetinib was resuspended in 0,5% hydroxypropylmethylcellulose (Sigma) and 0.4% Tween-80 in distilled water. All agents were administered via oral gavage.

### Statistical analysis

The results presented are representative of three independent experiments run in triplicate, unless otherwise indicated. Student's t test and two-way ANOVA were carried out using GraphPad Prism to calculate significance. Differences were considered significant at $p < 0.05$. Results are expressed as mean ± standard error (S.D.).

## Acknowledgements

We thank our colleagues for continuous support and discussions. CSV, IVK, and BP were supported by a 'Sta op tegen Kanker' International Translational Cancer Research Grant to HC and JLB. Stand Up to Cancer is a program of the Entertainment Industry Foundation administered by the AACR. HJS is supported by a KWF fellowship from the Dutch Cancer Society (UU 2013–6070) and JD by a VENI grant from the Netherlands Organization for Scientific Research (NWO-ZonMw) (91614138). The Mouse Clinic for Cancer and Aging (BvG, MvdV) is supported by National Roadmap grant for Large-Scale Research Facilities of the Netherlands Organization for Scientific Research. General support came from the Netherlands Organization for Scientific Research gravitation program Cancer Genomics Netherlands and the Josephine Nefkens Foundation.

## Additional information

### Competing interests

HC: An inventor on several patents involving the organoid culture system (USPTO 20120196312 and 20140256037). The other authors declare that no competing interests exist.

### Funding

| Funder | Author |
| --- | --- |
| Nederlandse Organisatie voor Wetenschappelijk Onderzoek | Jarno Drost |
| Stand Up To Cancer | Hans Clevers<br>Johannes L Bos |
| KWF Kankerbestrijding | Hugo J Snippert |

The funders had no role in study design, data collection and interpretation, or the decision to submit the work for publication.

### Author contributions

CSV, RMO, BP, SM, HJS, Conception and design, Acquisition of data, Analysis and interpretation of data, Drafting or revising the article; JD, HC, Conception and design, Drafting or revising the article, Contributed unpublished essential data or reagents; IV-K, Conception and design, Acquisition of data, Drafting or revising the article; BvG, MvdV, Designed and performed the in vivo drug response on the colorectal cancer organoids, Conception and design, Acquisition of data, Drafting or revising the article; MvdW, Conception and design, Analysis and interpretation of data, Drafting or revising the article, Contributed unpublished essential data or reagents; DAE, JLB, Conception and design, Analysis and interpretation of data, Drafting or revising the article; RB, Conception and design, Drafting or revising the article

### Ethics

Animal experimentation: Approval for this study was obtained by the local animal experimental committee at The Netherlands Cancer Institute (DEC-NKI). (DEC-NKI; OZP=12012 and WP5727 and WP5689). All of the animals were handled according to approved institutional animal care and use committee.

## Additional files

### Supplementary files

• Supplementary file 1. Dose-response data (%ATP normalized to DMSO) for all patient-derived tumor organoids as described in *Figure 4—source data 1*, *Figure 4—source data 2*, *Figure 5— source data 1* and *Figure 6—source data 1*.

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
