## [Decision Letter]

Thank you for submitting your article "Targeting mutant RAS in patient-derived colorectal cancer organoids by combinatorial drug screening" for consideration by *eLife*. Your article has been reviewed by three peer reviewers, and the evaluation has been overseen by a Reviewing Editor and Ivan Dikic as the Senior Editor. The following individual involved in review of your submission has agreed to reveal his identity: Michael Yaffe (Reviewer #3).

The reviewers have discussed the reviews with one another and the Reviewing Editor has drafted this decision to help you prepare a revised submission.

Summary:

This paper makes an important contribution to the emerging concept that CRC organoids are useful as a middle ground between 2-D cultures and mouse models. The authors use organoids with wildtype KRAS, organoids with mutant KRAS, and normal organoids engineered to express mutant KRAS, and find that KRAS mutational status determines the sensitivity to HER-RAS-ERK axis targeted therapies. While these observations had been previously described in vitro or in clinical trials, one new result is that MEKi/HERi or MEKi/ERKi combinations induce cell cycle arrest rather than cell death. Importantly, addition of BCLi during HERi/MEKi removal induces death. This could have clinical relevance given the platelet toxicity of BCLi. The paper demonstrates the potential of using patient-derived CRC organoids in testing and evaluating combinatorial drug responses.

Essential revisions:

The reviewers agree that the strength of the paper lies in the extensive data validation and use of patient-derived (and engineered) organoids. The chance of artifacts has been reduced by the use of varied inhibitors against common targets and the different approaches to measure cell death. However, the use of organoids for drug screens is not new, CRISPR-mediated mutagenesis of KRAS has been reported, and the efficacy of combinations of EGFRi and MEK/ERKi is well-established. Moreover, the reviewers were concerned that there is limited new biology or molecular mechanisms in the paper. For this reason, the reviewers are only willing to consider the paper for *eLife* if the new findings (of cell stasis rather than death following MEK/EGFRi and of the efficacy of BCLi) are better substantiated.

The reviewers have discussed their findings and agree that the following points must be addressed in a revised submission:

1) Organoids do not completely reflect the biology of cancer in vivo. The response of CRC organoids to combination therapy should be confirmed in in vivo models such as mouse flank xenograft of kidney capsule (e.g., Fujii et al. 2016 Cell Stem Cell 18:827).

2) The argument that EGFRi/MEKi and MEKi/ERKi cause cell cycle arrest rather than senescence or death is based on H2B fluorescence and a limited set of other assays. This evidence needs to be bolstered by independent assays, such as caspase-3, annexin V, cell cycle (p21), checkpoint, proliferation and senescence markers. The authors could sort for dead cells and then separate arrested cells by DNA content and identify arrested cells.

In addition, the authors should consider the other, less major, comments that are listed below.

*Reviewer #3:*

1) Figure 1, Panel B. A color key should be provided to understand the colored organoids, in order to easily identify a given z-stack depth with a particular color.

2) Figure 2, Panel B. The authors claim this full-matrix screen as good illustrator of synergy. Given that the heatmaps have been smoothened with walking (i.e. moving?) averages, it would be good to clarify better the concept.

3) Figure 2, Panel C. It looks like the introduction of mutant KRAS in P18T has an effect in cell proliferation even in absence of treatment (organoid size at 65h is clearly smaller that P18T one), which is not commented in the text. Is the data normalized to the starting number of cells?

4) Also, the authors claim the cells stop proliferating, but no additional validation than cell size is used (i.e. checkpoint activation, p21 expression).

5) Figure 4. The authors try to show that the response to targeted therapies is only dependent on KRAS status independently of cellular mutational status, as the results in normal organoids are comparable to cancer ones. I would be more cautious in this respect, as the culture of normal organoids requires high Wnt activity media, growth factors and inhibitors, which may not reflect the normal mucosa in the intestine. Additionally, I find kind of surprising that the introduction of an oncogenic KRAS does not result in oncogenic senescence in a normal organoid. It would be nice to see some organoid pictures in this set of experiments and/or a comparison of WT vs. KRAS-mutant organoids growth curves in absence or treatment.

6) Figure 5 find this figure as the major strength of the work, as it validates the organoid lines screening as a real tool to assess drug/drug combination sensitivity. The data here shows a pretty wide range of sensitivities to afatinib alone for both the wt and mutant K-Ras organoids, as the authors note in the subsection “Therapy-surviving cancer cells quickly restart tumor growth after release of targeted inhibition”, second paragraph. Can the authors use the genotypic information they have on the organoids to elucidate any other contributors to the relative monotherapy resistance?

7) Figure 6. The authors describe here a novel and smart strategy to track organoid viability in real-time cultures, and developed ImageJ macros to quantify the phenotypes, that might be really useful for other researchers performing 3D cultures (organoids, spheroids, etc.). Nevertheless, they should clarify in the figure legend that cell dead/alive status is identified by nuclear size (red colored nuclei may be associated to π staining, also described in the Methods section and used in following experiments in the manuscript). Also, given the novelty of the method, I would recommend further validation of nuclear size as a proxy of dead/alive status (co-stain with apoptotic markers and/or viability dyes).

8) The authors conclude that the inhibition of the EGFR/MEK/ERK axis induces mostly a cell cycle arrest that, in absence of drug, can be reverted and cells proliferate again. I would be more cautious in this affirmation, given the data presented. It is not clear from the data presented whether there is a global cell cycle stop (no assessment of G1/S checkpoint activation nor cell cycle progression markers) and the data does not rule out the possibility of a small population of pre-resistant cells within the organoid, which are then able to repopulate in absence of treatment.

9) Figure 6, Panel B. – Please add a color code legend.

10) Figure 7, Panel A. I would suggest splitting this graph in at least three different ones, in order to better point out the synergy between targeted drugs and navitoclax. Panel B. PI final concentration is not provided in the methods section, which may impact the threshold applied for the analysis (i.e. π unspecific staining/background). Panel C. The authors claim they need a robust inhibition of both MEK and ERK in order to Navitoclax to be effective at low concentrations. It would be very helpful, to show some western blots of p-EGFR, p-ERK and downstream signaling to demonstrate such inhibition at 1uM of afatinib and selumetinib vs. lower concentrations of them (i.e. 0.065uM).

11) Figure 3—figure supplement 1 – what is the rectangle code for the inhibitors?

[Editors' note: further revisions were requested prior to acceptance, as described below.]

Thank you for resubmitting your work entitled "Targeting mutant RAS in patient-derived colorectal cancer organoids by combinatorial drug screening" for further consideration at *eLife*. Your revised article has been favorably evaluated by Ivan Dikic (Senior editor), a Reviewing editor, and two reviewers.

The reviewers are now very positive about the paper, but there are some remaining concerns. New experiments are not required, but you may have the data and could add them or discuss. Points 1-3 address whether growth rate differences between WT and mutant KRAS organoids might be detectable in low serum/glucose conditions and whether EGFRi and MEKi are really inhibiting as expected in vivo. Point 4 suggests adding in vivo histology to complement the in vitro histology of the organoids. Point 5 concerns whether the in vivo resistance is due to selection. Point 6 requests statistics for the growth curves and cell cycle and this point must be addressed. Point 7 suggests adding comments about Bcl2 vs. BclXL drugs.

1) The authors write that "Upon addition of oncogenic KRAS, no overall differences in morphology or growth rates were observed." (Results). It has been established that Kras mutant cell lines exhibit greater proliferation than Kras wild-type cell lines in low serum or low glucose conditions, and that tumors in Kras genetically engineered mouse models also grow faster than Kras wild-type GEMMs (Yun et al. Glucose deprivation contributes to the development of KRAS pathway mutations in tumor cells Science 2009 and Martin et al. Development of a colon cancer GEMM-derived orthotopic transplant model for drug discovery and validation Clinical Cancer Research 2013). Therefore, it will be of interest to the readership if the authors could assess using CRISPR engineered and/or human CRC organoids how KRAS mutant vs. wild-type organoids grow in vitro under low serum or low glucose conditions, and/or in vivo upon transplantation.

2) Figure 3—figure supplement 3. The authors performed western blots on Kras mut and WT organoids in response to drug treatment, as requested. The lower P-ERK level in control KRAS MUT organoids vs. control WT organoids may an artifact of the media conditions. The authors should assess P-ERK and/or P-MEK in Low serum / low glucose conditions.

3) Figure 2 and Figure 2—figure supplement 1. Western blots for P-ERK and/or P-MEK should be provided for in vivo experiments to demonstrate that tumor growth / stasis is due to the mechanism of action of the drugs. This would easily address the authors' point that they "speculate that the in vivo drug concentrations were insufficient to effectively block the EGFR-MEK-ERK pathway" for the low dose combination treatment.

4) Please provide in vivo histology of the tumors compared to the in vitro organoids, with or without drug treatment.

5) It would have been more interesting to me to see secondary in vitro passage of the residual in vivo tumors after HIGH dose treatment. This would answer the question of whether a therapeutically relevant drug dose reveals primary resistance or induces secondary resistance in the tumor organoid tissue. This is a comment rather than a request for more experiments – may be more of a topic for future study.

6) The drug dose response curves and cell cycle data do not indicate significant differences or appropriate statistics in the figures nor in the legends. This should be addressed.

7) Regarding the naviticlax combination. AbVie has drugs that now target either Bcl2 or BclXL so future experiments could investigate the effects of separate drugs or knockdown and then co-treatment with RAS pathway targeting drugs.

Figure 3—figure supplement 3: phosphorylation is misspelled.

---

## [Author Response]

[…]

*Essential revisions:*

*The reviewers agree that the strength of the paper lies in the extensive data validation and use of patient-derived (and engineered) organoids. The chance of artifacts has been reduced by the use of varied inhibitors against common targets and the different approaches to measure cell death. However, the use of organoids for drug screens is not new, CRISPR-mediated mutagenesis of KRAS has been reported, and the efficacy of combinations of EGFRi and MEK/ERKi is well-established. Moreover, the reviewers were concerned that there is limited new biology or molecular mechanisms in the paper. For this reason, the reviewers are only willing to consider the paper for eLife if the new findings (of cell stasis rather than death following MEK/EGFRi and of the efficacy of BCLi) are better substantiated.*

*The reviewers have discussed their findings and agree that the following points must be addressed in a revised submission:*

*1) Organoids do not completely reflect the biology of cancer in vivo. The response of CRC organoids to combination therapy should be confirmed in in vivo models such as mouse flank xenograft of kidney capsule (e.g., Fujii et al. 2016 Cell Stem Cell 18:827).*

We addressed the in vivo drug response of colorectal cancer organoids (new Figure 2). Most notably, we have xenotransplanted KRAS mutant P26T whose in vivo drug responses phenocopy the ones we measured in vitro; i.e. upon medium dosage of EGFRi + MEKi there is limited effect, while at the highest dosage we observe stalling of tumor growth, indicating loss of proliferative activity. Moreover, the in vivo drug response (growth stabilization, but no regression) was similar to previous reports of drug response on PDX models of RAS mutant CRC (Sun et al., 2014). By our knowledge, this is the first report that compared drug response of tumor organoids between in vivo and in vitro culture conditions. We agree with the reviewers that this is a very important addition to the manuscript as it underscores the power of tumor organoid libraries to facilitate pre-screening of new drug candidates and drug combinations prior to resource-intensive mouse experiments and clinical trials.

We included the following addition to the Results section:

*“*in vivodrug response of xenotranplanted patient-derived cancer organoids

In order to validate the observed drug response of in vitro cultured organoids in an in vivo model, we xenotransplanted P18T and P26T tumor organoids in immunodeficient mice. […] The fact that in vivo xenografted CRC organoids yields similar drug responses as in vitro organoid cultures and identical to previous reported drug response of KRAS mutant PDX models of CRC (Sun et al., 2014), validates the testing and evaluation of targeted inhibitors in CRC organoids.”

Currently, we are still optimizing reliable engraftment of P8T or P18T tumor organoids as positive controls for EGFRi therapies, in line with a previous publication regarding unsuccessful engraftments of tumor organoids with lower number of cancer mutations (Drost et al., 2015).

Moreover, we have performed in vivo drug responses using the addition of navitoclax (BCLi) to EGFRi/MEKi combination as well. As expected, low concentration navitoclax (50mg/kg) was well tolerated by the mice, but addition of low concentration of navitoclax to EGFRi/MEKi as a triple combination turned out to be too toxic for the mice in order to complete the experiments.

We discuss above-mentioned results in the manuscript:

*“*With respect to clinical applications, we here report that effective inhibition of the EGFR-MEK-ERK pathway through combinatorial targeting does significantly prime the cytostatic RAS mutant cancer cells for apoptosis. […] Nevertheless, we consider the navitoclax-induced apoptosis as a proof-of-principle that EGFR-MEK-ERK pathway inhibition in combination with alternative signaling nodes holds great promise in identifying therapeutic drug combinations that kill RAS mutant tumor cells while being tolerated by the patient.”

Importantly, we re-established organoids from the different treated tumors in mice. Subsequent drug responses of these secondary post-xenograft organoids were unaltered in comparison with the parental tumor organoids that were xenotransplanted, suggesting that tumor phenotype regarding drug response remains constant independent of in vitro culture or upon xenotransplantation.

We included the following addition to the manuscript:

“To exclude that the tumors had acquired resistance during the in vivo drug treatment, we isolated the tumors to re-establish secondary organoids and subjected these to identical drug tests. […] Indeed, in agreement with lower drug concentrations that proved to be ineffective in blocking proliferation in vitro (Figure 1—figure supplement 1, Video 1), we speculate that the in vivo drug concentrations were insufficient to effectively block the EGFR-MEK-ERK pathway.”

*2) The argument that EGFRi/MEKi and MEKi/ERKi cause cell cycle arrest rather than senescence or death is based on H2B fluorescence and a limited set of other assays. This evidence needs to be bolstered by independent assays, such as caspase-3, annexin V, cell cycle (p21), checkpoint, proliferation and senescence markers. The authors could sort for dead cells and then separate arrested cells by DNA content and identify arrested cells.*

We strengthened our evidence that KRAS mutant tumor cells are blocked in cell cycle progression upon inhibition of the EGFR-MEK-ERK pathway. These data are now included as a new Figure 8.

First, we performed cell cycle analysis by flow cytometry. Most notably, after 72hrs of drug treatment we observed that most cancer cells accumulate in G1 at the expense of tumor cells in S-phase. Indeed, this is in agreement with our live-cell imaging data of the drug response, where virtual no mitotic events were detected towards the end of the treatment.

We included the following addition to the manuscript:

“Dual inhibition of the EGFR-MEK-ERK pathway induces a G1 cell cycle arrest

To characterize the induced cell cycle arrest in RAS mutant tumor cells in more detail, we performed cell cycle analysis by flow cytometry using a 3hr EdU pulse in combination with DNA staining to discriminate between the different cell cycle phases (G1, S and G2 respectively) in P18T-KRASG12D and P26T. Indicative of a G1 arrest, we observed a sharp decline in the amount of RAS mutant tumor cells in S-phase using inhibitor combinations EGFRi/MEKi and MEKi/ERKi, while a similar fraction of cells accumulated in G1 (Figure 8).”

Subsequently, using functional assays we addressed whether the vast majority of tumor cells re-obtained proliferative activity after drug withdrawal, in contrast to senescence or a minor subpopulation being responsible for tumor relapse. First, we performed EdU incorporations at various time points during the drug response and drug recovery. In agreement with the inhibition of cell cycle progression, there is hardly any EdU incorporation in the presence of the drugs (no cells in S-phase), while the vast majority of the tumor cells incorporated EdU during the first 3 days after drug withdrawal suggesting renewed proliferative activity in virtually all tumor cells.

Second, we performed real-time imaging of tumor organoids after drug withdrawal (drug recovery) and we quantified the number of mitotic and apoptotic events over time. Indeed, starting at ~20hrs after withdrawal of the drugs, we observed frequent mitotic events again.

We included the following addition to the manuscript:

“To further characterize drug-induced arrest, we investigated whether the regrowth after drugs washout involves all therapy-surviving tumor cells or only a minor subpopulation that fuels tumor relapse. For this, we performed two functional assays. […] Indeed, in line with a G1 arrest, we observed a delay of about 20-24hrs after withdrawal of the drugs (EGFRi/MEKi) before observing numerous mitotic events again in all regions of the arrested organoids (Figure 8). (Similar results were obtained for MEKi/ERKi, data not shown).”

*In addition, the authors should consider the other, less major, comments that are listed below.*

*Reviewer #3:*

*1) Figure 1, Panel B. A color key should be provided to understand the colored organoids, in order to easily identify a given z-stack depth with a particular color.*

Done.

*2) Figure 2, Panel B. The authors claim this full-matrix screen as good illustrator of synergy. Given that the heatmaps have been smoothened with walking (i.e. moving?) averages, it would be good to clarify better the concept.*

We agree with the referee that we should have better clarified the concept. To indicate for which concentration ratios the effects are larger than expected, we now provided heat maps with calculated ‘Bliss scores’ where positive scores indicate combinations where the effect is greater than additive.

We made the following addition to the manuscript and included an extra supplemental figure:

“Moreover, we analyzed combination effects using the Bliss independence model. Positive Bliss scores indicate combinatorial effects that exceed additive effects. The heat map of Bliss scores for P18T and P18T-KRAS show that a large range of concentrations for both compounds show positive scores, but that presence of oncogenic KRAS renders the loss of viability and positive Bliss range towards higher drug concentrations indicating resistance (Figure 3—figure supplement 1).”

*3) Figure 2, Panel C. It looks like the introduction of mutant KRAS in P18T has an effect in cell proliferation even in absence of treatment (organoid size at 65h is clearly smaller that P18T one), which is not commented in the text. Is the data normalized to the starting number of cells?*

The depicted movie is just an example representing several organoids. In general, for all drug responses that were measured, either by ATP assays or by real-time imaging, the size and growth of organoids were synchronized by tripsinization 5 days prior drug response and size filtering just prior drug response. Although variability between individual organoids existed, overall no differences in growth speed were observed.

We made the following addition to the manuscript:

“Upon addition of oncogenic KRAS, no overall differences in morphology or growth rates were observed.”

Moreover, we quantified mitotic and apoptotic events during the 72hrs live-cell imaging of the drug response over multiple organoids and added panel D to Figure 3 and Figure 3—figure supplement 2:

“More specifically, quantifications of all mitotic and apoptotic events during the filmed drug response revealed both loss of proliferation and apoptosis induction in P18T, while P18T-KRASG12D only showed reduced proliferation but unchanged apoptotis rates (Figure 3 and Figure 3—figure supplement 2).”

*4) Also, the authors claim the cells stop proliferating, but no additional validation than cell size is used (i.e. checkpoint activation, p21 expression).*

We indeed claimed that cells stop proliferating, which we mainly based on the live-cell imaging data of various drug responses where only relatively few mitotic events were observed during the 2^nd^ and 3^rd^ day of the drug response (Video 1).

In short: We now substantiated the evidence that KRAS mutant cells loose proliferative activity upon combination therapy against the EGFR-MEK-ERK pathway. Please see rebuttal regarding the second main point for comments and additions to the manuscript.

*5) Figure 4. The authors try to show that the response to targeted therapies is only dependent on KRAS status independently of cellular mutational status, as the results in normal organoids are comparable to cancer ones. I would be more cautious in this respect, as the culture of normal organoids requires high Wnt activity media, growth factors and inhibitors, which may not reflect the normal mucosa in the intestine. Additionally, I find kind of surprising that the introduction of an oncogenic KRAS does not result in oncogenic senescence in a normal organoid. It would be nice to see some organoid pictures in this set of experiments and/or a comparison of WT vs. KRAS-mutant organoids growth curves in absence or treatment.*

“This may imply that the sensitivity of RAS WT colon cancer for EGFRi is not an acquired oncogene-addiction, but merely represents the dependency of normal colon (stem) cells on EGFR signaling activity (Wong et al., 2012). Indeed, normal organoids from the human colon consist predominantly of proliferative stem and progenitor cells due to the WNT ligands in the culture medium. Therefore, the observed toxicity in the normal organoids may very well be most representative for direct effects on the stem cell compartments of the normal colon. Indeed, one of the direct side effects of anti-EGFR monotherapy is diarrhea (Miroddi et al., 2015).”

Additionally, I find kind of surprising that the introduction of an oncogenic KRAS does not result in oncogenic senescence in a normal organoid. It would be nice to see some organoid pictures in this set of experiments and/or a comparison of WT vs. KRAS-mutant organoids growth curves in absence or treatment.

In contrast to overexpression of oncogenic KRAS, activation of oncogenic KRAS from its endogenous locus in mouse colon does not lead to senescence (Snippert et al., EMBO rep 2014). In that respect, our organoid models where we used CRISPR/Cas9-mediated homologous recombination to introduce oncogenic KRAS at its endogenous locus are in agreement with mouse studies.

In order to strengthen the manuscript regarding this point, we have now included an extra supplemental figure comparing morphology and cell cycle of normal organoids and normal organoid with KRAS^G12D^. Moreover, we have added the following statement in the manuscript:

“In analogy with mouse studies (Snippert et al., 2014), we observed no morphological alteration nor induction of senescence upon introduction of oncogenic KRAS (Figure 5—figure supplement 1).”

*6) Figure 5 find this figure as the major strength of the work, as it validates the organoid lines screening as a real tool to assess drug/drug combination sensitivity. The data here shows a pretty wide range of sensitivities to afatinib alone for both the wt and mutant K-Ras organoids, as the authors note in the subsection “Therapy-surviving cancer cells quickly restart tumor growth after release of targeted inhibition”, second paragraph. Can the authors use the genotypic information they have on the organoids to elucidate any other contributors to the relative monotherapy resistance?*

This is a very interesting remark of the referee. As indicated, we have been able to use the genetic information of the organoids to understand the most dramatic ‘KRASwt’ outliers, i.e. resistant organoid lines P19T and P25T respectively. Both turned out to contain alternative well-known oncogenic mutations in the downstream EGFR signaling pathway, i.e. BRAF (V600E) and NRAS (Q61H). It will be much more challenging in order to find mutations that do not confer complete resistance such as oncogenic KRAS, NRAS or BRAF, but do contribute to such a phenotype. We therefore hesitate to speculate about possible resistance mechanisms imposed by low frequency cancer mutations due to the relative small sample size of organoids. Indeed, we do agree that mechanistic studies on these lower abundant mutation frequencies in relation to drug sensitivity are very interesting, but we consider it beyond the scope of this manuscript.

*7) Figure 6. The authors describe here a novel and smart strategy to track organoid viability in real-time cultures, and developed ImageJ macros to quantify the phenotypes, that might be really useful for other researchers performing 3D cultures (organoids, spheroids, etc.). Nevertheless, they should clarify in the figure legend that cell dead/alive status is identified by nuclear size (red colored nuclei may be associated to PI staining, also described in the Methods section and used in following experiments in the manuscript).*

We have now clearly indicated in the figure legend that cell dead/alive status is identified by nuclear size.

“Live nuclei and dead nuclear remnants were marked for each z-plane, as identified by nuclear size (3rd panel, see methods section), measured …”

*Also, given the novelty of the method, I would recommend further validation of nuclear size as a proxy of dead/alive status (co-stain with apoptotic markers and/or viability dyes).*

Well taken. We have now analyzed a specially acquired training set to demonstrate the feasibility of our ImageJ scripts. We have now included this analysis as a new figure (Figure 7—figure supplement 1) to which we refer from the Methods section

“Of note, this method is independent of non-permeable DNA dyes such as PI to avoid their potential long-term effect on organoid growth. In order to validate the current method, a single time point data set was acquired with the use of PI (see Figure 7—figure supplement 1), validating the robustness of the strategy.”

“Figure 7—figure supplement 1

The custom-made image analysis software for quantifications in Figure 7 is extensively described in the Methods section. […] B) 16 z-stacks were analyzed by the two analysis methods (see A) to yield surface ratios (dead/total H2B), that show strong correlation (Pearsson coefficient of 0.76, p<0.001).”

*8) The authors conclude that the inhibition of the EGFR/MEK/ERK axis induces mostly a cell cycle arrest that, in absence of drug, can be reverted and cells proliferate again. I would be more cautious in this affirmation, given the data presented. It is not clear from the data presented whether there is a global cell cycle stop (no assessment of G1/S checkpoint activation nor cell cycle progression markers) and the data does not rule out the possibility of a small population of pre-resistant cells within the organoid, which are then able to repopulate in absence of treatment.*

Most importantly, in order to discriminate between a small population of pre-resistant cells versus all cells that contribute to tumor growth in the absence of treatment, we have now assessed the loss and gain of proliferative activity using two functional assays.

First, we performed EdU incorporation at various time points during the recovery phase after the drug withdrawal. Second, we performed real-time imaging of the organoid recovery after drug withdrawal. For detailed discussion, comments and manuscript additions, please see our rebuttal of the second main point.

*9) Figure 6, Panel B. Please add a color code legend.*

The color code legend in panel C was indicative for panel B as well. We have now clearly stated this in the legend of panel B.

“Color code legend is provided at the bottom of panel C.”

*10) Figure 7, Panel A. I would suggest splitting this graph in at least three different ones, in order to better point out the synergy between targeted drugs and navitoclax.*

We have now split the graph (now Figure 9) into three different ones to optimize readability.

*Panel B. PI final concentration is not provided in the methods section, which may impact the threshold applied for the analysis (i.e. PI unspecific staining/background).*

We have now provided the concentration in the Methods section.

*Panel C. The authors claim they need a robust inhibition of both MEK and ERK in order to Navitoclax to be effective at low concentrations. It would be very helpful, to show some western blots of p-EGFR, p-ERK and downstream signaling to demonstrate such inhibition at 1uM of afatinib and selumetinib vs. lower concentrations of them (i.e. 0.065uM).*

We have now performed Western blot analysis of p-ERK inhibition in RAS mutant tumor organoids P18T-KRAS^G12D^ and P26T using different concentrations of afatinib (EGFRi) and selumetinib (MEKi). Indeed only high concentrations of both drugs are able to significantly suppress p-ERK activity, confirming the notion that minimal levels of ERK activity (i.e. high concentrations and/ or dual targeting) are required in order for low amounts of navitoclax to be effective. We have included the new data in Figure 9—figure supplement 3 and referenced it in the text.

*“*In light of the dose-limiting toxicity of navitoclax in blood platelets, we performed full matrix-screens to explore optimal combinations of drug concentrations (Figure 9). In agreement with previous results, the more efficient inhibition of the RAS pathway, i.e. high concentrations and/or dual targeting (Figure 9—figure supplement 3) the lower the concentration of navitoclax that is necessary to affect cellular viability.*”*

*11) Figure 3—figure supplement 1 – what is the rectangle code for the inhibitors?*

Our apologies. We have now included a legend for the inhibitors.

[Editors' note: further revisions were requested prior to acceptance, as described below.]

[…]

*The reviewers are now very positive about the paper, but there are some remaining concerns. New experiments are not required, but you may have the data and could add them or discuss. Points 1-3 address whether growth rate differences between WT and mutant KRAS organoids might be detectable in low serum/glucose conditions and whether EGFRi and MEKi are really inhibiting as expected in vivo. Point 4 suggests adding in vivo histology to complement the in vitro histology of the organoids. Point 5 concerns whether the in vivo resistance is due to selection. Point 6 requests statistics for the growth curves and cell cycle and this point must be addressed. Point 7 suggests adding comments about Bcl2 vs. BclXL drugs.*

*1) The authors write that "Upon addition of oncogenic KRAS, no overall differences in morphology or growth rates were observed." (Results). It has been established that Kras mutant cell lines exhibit greater proliferation than Kras wild-type cell lines in low serum or low glucose conditions, and that tumors in Kras genetically engineered mouse models also grow faster than Kras wild-type GEMMs (Yun et al. Glucose deprivation contributes to the development of KRAS pathway mutations in tumor cells Science 2009 and Martin et al. Development of a colon cancer GEMM-derived orthotopic transplant model for drug discovery and validation Clinical Cancer Research 2013). Therefore, it will be of interest to the readership if the authors could assess using CRISPR engineered and/or human CRC organoids how KRAS mutant vs. wild-type organoids grow in vitro under low serum or low glucose conditions, and/or in vivo upon transplantation.*

At the moment, we haven’t studied morphology and proliferation in alternative culture conditions. Organoid culture conditions are determined per tissue type and optimized for normal organoids that are derived from healthy tissue. Normally, we strive to culture all derivatives of normal organoids (CRISPR engineered tumor lines) and/or patient-derived tumor organoids in identical growth conditions.

To avoid confusion, we changed the manuscript text to explicitly state that we only studied morphology and proliferation under normal culture conditions.

“[…]were observed during normal culture conditions.”

*2) Figure 3—figure supplement 3. The authors performed western blots on Kras mut and WT organoids in response to drug treatment, as requested. The lower P-ERK level in control KRAS MUT organoids vs. control WT organoids may an artifact of the media conditions. The authors should assess P-ERK and/or P-MEK in Low serum / low glucose conditions.*

We agree that it might be interesting to examine P-MEK and P-ERK levels under different conditions, but for this paper the in vitro conditions used for the screens are in our opinion the most appropriate (see also comments in 1).

*3) Figure 2 and Figure 2—figure supplement 1. Western blots for P-ERK and/or P-MEK should be provided for in vivo experiments to demonstrate that tumor growth / stasis is due to the mechanism of action of the drugs. This would easily address the authors' point that they "speculate that the in vivo drug concentrations were insufficient to effectively block the EGFR-MEK-ERK pathway" for the low dose combination treatment.*

We agree with the reviewers that the suggested experiments in point 3, as well as in 4 and 5 would be nice additions, but in our opinion they are not crucial for the conclusions reached in this paper. A thorough analysis of these in vivo tumor characteristics, however, will be the subject of future experiments.

*4) Please provide in vivo histology of the tumors compared to the in vitro organoids, with or without drug treatment.*

*5) It would have been more interesting to me to see secondary in vitro passage of the residual in vivo tumors after HIGH dose treatment. This would answer the question of whether a therapeutically relevant drug dose reveals primary resistance or induces secondary resistance in the tumor organoid tissue. This is a comment rather than a request for more experiments – may be more of a topic for future study.*

*6) The drug dose response curves and cell cycle data do not indicate significant differences or appropriate statistics in the figures nor in the legends. This should be addressed.*

Well taken. We have now included statistics with respect to Figure 2 (in vivo drug response) and Figure 8 (cell cycle analysis).

For Figure 2 we have calculated the statistical significant differences between the growth rates of the tumors. Statistics is now indicated in the figure with *, included in the figure legends and in the manuscript.

“[…]but we observed no significant effect of the …”.

“[…]indeed induced significant growth stabilization …”.

For Figure 8 we have calculated the statistical significance that dual inhibition of the EGFR-MEK-ERK pathway significantly changes the cell cycle distribution (Figure 8). We have included these statements in the figure legend of Figure 8.

“Dual inhibition of the EGFR-MEK-ERK pathway significantly changes the distribution of cells between stages of the cell cycle (Chi^2^: all p values < 0,0001) with[…]”.

In addition, we have determined the average growth rates of the tumor organoids during recovery of drug treatment (Figure 8), which we included in a new supplemental figure (Figure 8—figure supplement 1).

Subsection “Dual inhibition of the EGFR-MEK-ERK pathway induces a G1 cell cycle arrest”, last paragraph and legend to Figure 8—figure supplement 1. “Average growth speeds of the organoids were determined by linear fitting of the traces shown in Figure 8. The two time frames roughly correspond to the first half and the second half of the experiment. Directly after drug removal, afatinib- and selumetinib-treated organoids show a significant reduction in growth speed as compared to vehicle-treated organoids. After 22-24 hrs of recovery, growth rates return to the level of vehicle-treated organoids (possibly even slightly faster). *, p<0,05; **, p<0,01; ***, p<0,001; n.s., not significant.”

*7) Regarding the naviticlax combination. AbVie has drugs that now target either Bcl2 or BclXL so future experiments could investigate the effects of separate drugs or knockdown and then co-treatment with RAS pathway targeting drugs.*

Indeed, we are very keen to perform these sorts of experiment in the near future. However, in the current study we restricted ourselves to small molecule inhibitors that are already in advanced stages of different clinical trials, while A-1155463 (BCLXL specific lead compound from Abvie) is still for research purposes only.

*Figure 3—figure supplement 3: phosphorylation is misspelled.*

Corrected.